# Regime-Adaptive Bayesian Optimization via Dirichlet Process Mixtures of Gaussian Processes

**Yan Zhang** [1]   **Xuefeng Liu** [2]   **Sipeng Chen** [1]   **Sascha Ranftl** [3]   **Chong Liu** [4]   **Shibo Li** [1]

## Abstract

Standard Bayesian Optimization (BO) assumes uniform smoothness across the search space—an assumption violated in multi-regime problems such as molecular conformation search through distinct energy basins or drug discovery across heterogeneous molecular scaffolds. A single GP either oversmooths sharp transitions or hallucinates noise in smooth regions, yielding miscalibrated uncertainty. We propose RAMBO, a Dirichlet Process Mixture of Gaussian Processes that automatically discovers latent regimes during optimization, each modeled by an independent GP with locally-optimized hyperparameters. We derive collapsed Gibbs sampling that analytically marginalizes latent functions for efficient inference, and introduce adaptive concentration parameter scheduling for coarse-to-fine regime discovery. Our acquisition functions decompose uncertainty into intra-regime and inter-regime components. Experiments on synthetic benchmarks and real-world applications—including molecular conformer optimization, virtual screening for drug discovery, and fusion reactor design—demonstrate consistent improvements over state-of-the-art baselines on multi-regime objectives. Code is available at https://github.com/AnthonyZhangYan/RAMBO.

## 1 Introduction

Bayesian Optimization (BO) has become the standard approach for optimizing expensive black-box functions, with applications spanning hyperparameter tuning (Snoek et al., 2012; Feurer & Hutter, 2019), neural architecture search (Zoph & Le, 2017; Elsken et al., 2019; Kandasamy et al., 2018), materials discovery (Lookman et al., 2019; Xue et al., 2016; Kusne et al., 2020), and drug design (Gómez-Bombarelli et al., 2018; Sanchez-Lengeling & Aspuru-Guzik, 2018). By fitting a Gaussian Process (GP) surrogate to observed data $\mathcal{D}$ (Rasmussen, 2003) and using acquisition functions to guide sampling (Jones et al., 1998; Mockus, 1998), BO efficiently navigates high-dimensional spaces with minimal function evaluations.

However, standard BO with stationary kernels assumes uniform smoothness and noise characteristics across the search space. While non-stationary kernel constructions exist—such as input-dependent length scales (Paciorek & Schervish, 2003; Higdon et al., 1999; Plagemann et al., 2008) or deep GP compositions (Damianou & Lawrence, 2013; Wilson et al., 2016a)—they model smoothly-varying hyperparameters and require specifying the functional form of this variation a priori. In contrast, many scientific design problems exhibit discrete regime structure with abrupt transitions rather than gradual parameter drift. In molecular conformation search (Hawkins et al., 2010; Riniker & Landrum, 2015), rotatable bonds create distinct energy basins separated by torsional barriers—each basin locally smooth, but the global landscape comprising hundreds of such basins with incommensurable curvature. Drug discovery landscapes (Gómez-Bombarelli et al., 2018; Griffiths & Hernández-Lobato, 2020; Korovina et al., 2020) are fragmented across molecular scaffolds, where different chemical families exhibit fundamentally different structure-activity relationships. Fusion reactor design (Gates et al., 2018; Cadena et al., 2025) traverses qualitatively different stability regimes as plasma geometry varies. In each domain, the objective function is not a smooth surface with slowly-varying properties but a patchwork of locally coherent regions separated by sharp boundaries. This discrete heterogeneity is poorly captured by continuous non-stationary kernels, which must interpolate smoothly between regimes and cannot represent the abrupt transitions that characterize real scientific landscapes. A mixture model, by contrast, naturally represents this structure: each component captures a distinct regime with its own hyperparameters, while the probabilistic

---

[1]Department of Computer Science, Florida State University, Tallahassee, FL, USA [2]School of Medicine, Stanford University, Stanford, CA, USA [3]School of Mechanical Engineering, Purdue University, West Lafayette, IN, USA [4]Department of Computer Science, University at Albany, State University of New York, Albany, NY, USA. Correspondence to: Shibo Li <shiboli@cs.fsu.edu>.

*Proceedings of the 43rd International Conference on Machine Learning*, Seoul, South Korea. PMLR 306, 2026. Copyright 2026 by the author(s).

assignment mechanism identifies regime boundaries directly from data without requiring their functional form to be specified in advance.

Building on this insight, we propose Regime-Adaptive Mixture Bayesian Optimization (RAMBO), which replaces the monolithic GP surrogate with a Dirichlet Process Mixture Model of Gaussian Processes (DPMM-GP). This nonparametric Bayesian framework adaptively partitions the search space into an unknown number of regimes inferred directly from data, with each regime modeled by an independent GP with locally-optimized hyperparameters. The Dirichlet Process prior provides automatic model selection: given $n$ observations, the expected number of discovered regimes $K$ grows as $\mathbb{E}[K \mid n] \approx \alpha \log(n/\alpha + 1)$ (Antoniak, 1974), adapting complexity to data without manual specification. We further introduce **adaptive concentration parameter scheduling** to control regime discovery dynamics throughout optimization. The concentration parameter $\alpha$ governs the propensity to create new regimes, and its optimal value varies with the amount of available data. Early stages benefit from small $\alpha$ to avoid premature fragmentation when observations are sparse; later stages permit larger $\alpha$ to discover fine-grained structure as evidence accumulates. This scheduling mirrors the exploration-exploitation tradeoff inherent to BO, but operates at the model complexity level rather than the sampling location level. Our contributions are as follows:

- We develop a complete DPMM-GP surrogate for BO with collapsed Gibbs sampling that analytically marginalizes latent functions, improving mixing efficiency over HMC-based inference (Rasmussen & Ghahramani, 2001).

- We introduce adaptive $\alpha$-scheduling to dynamically adjust model complexity—starting with small $\alpha$ to avoid premature fragmentation, then increasing to enable fine-grained regime discovery as data accumulates.

- We derive closed-form Expected Improvement for the DPMM-GP posterior, naturally decomposing uncertainty into intra-regime variance and inter-regime disagreement.

- We conduct extensive experiments on synthetic benchmarks and scientific applications—molecular conformer optimization, virtual screening for drug discovery, and fusion reactor design—demonstrating consistent improvements over state-of-the-art baselines on multi-regime objectives.

**Conflict of Interest Disclosure.**   The authors declare no financial conflicts of interest. None of the empirical evaluations in this paper involves a model, system, or commercial product developed by an entity employing any of the authors. The benchmark datasets used (ConStellaration, ZINC15/6T2W docking scores) are publicly released by third parties, and all baselines are run using their official

open-source implementations cited in Section 6.

## 2   Background

**Gaussian Processes.**   A Gaussian Process (GP) defines a distribution over functions $f(\mathbf{x})$ (Rasmussen, 2003; Williams & Rasmussen, 1995; MacKay, 1998), fully specified by a mean function $m(\mathbf{x})$ and covariance (kernel) function $k(\mathbf{x}, \mathbf{x}')$, written $f(\mathbf{x}) \sim \mathcal{GP}(m(\mathbf{x}), k(\mathbf{x}, \mathbf{x}'))$. In BO, we typically assume a zero-mean prior and use stationary kernels (Genton, 2001; Schölkopf & Smola, 2002) such as the Squared Exponential (SE): $k(\mathbf{x}, \mathbf{x}') = \sigma_f^2 \exp\left(-\frac{\|\mathbf{x}-\mathbf{x}'\|^2}{2\ell^2}\right)$, where $\sigma_f^2$ is the signal variance and $\ell$ is the length scale controlling smoothness or Matérn (Matérn, 1960).   Given observations $\mathcal{D}_n = \{(\mathbf{x}_i, y_i)\}_{i=1}^n$ with $y_i = f(\mathbf{x}_i) + \epsilon_i$ and $\epsilon_i \sim \mathcal{N}(0, \sigma_n^2)$, the posterior at a test point $\mathbf{x}_*$ is Gaussian: $p(f(\mathbf{x}_*) \mid \mathcal{D}_n) = \mathcal{N}(\mu_n(\mathbf{x}_*), \sigma_n^2(\mathbf{x}_*))$, with $\mu_n(\mathbf{x}_*) = \mathbf{k}_*^\top(\mathbf{K} + \sigma_n^2\mathbf{I})^{-1}\mathbf{y}; \sigma_n^2(\mathbf{x}_*) = k(\mathbf{x}_*, \mathbf{x}_*) - \mathbf{k}_*^\top(\mathbf{K}+\sigma_n^2\mathbf{I})^{-1}\mathbf{k}_*$, where $\mathbf{K}$ is the kernel matrix with $K_{ij} = k(\mathbf{x}_i, \mathbf{x}_j)$ and $\mathbf{k}_* = [k(\mathbf{x}_1, \mathbf{x}_*), \ldots, k(\mathbf{x}_n, \mathbf{x}_*)]^\top$. Computing the matrix inverse requires $\mathcal{O}(n^3)$ operations, but this is tractable in BO where $n$ is typically small. The critical limitation is that the hyperparameters $\theta = \{\ell, \sigma_f^2, \sigma_n^2\}$ are global: if the function varies rapidly in one region and slowly in another, a single GP estimates a compromise length scale that performs poorly in both.

**Bayesian Optimization.**   BO (Mockus, 1998; Brochu et al., 2010; Shahriari et al., 2016; Frazier, 2018) seeks the global optimum of an expensive black-box function: $\mathbf{x}^* = \operatorname{argmax}_{\mathbf{x} \in \mathcal{X}} f(\mathbf{x})$, where $f : \mathcal{X} \to \mathbb{R}$ is costly to evaluate, $\mathcal{X} \subseteq \mathbb{R}^d$ is a compact domain, and observations are noisy: $y = f(\mathbf{x}) + \epsilon$, where $\epsilon \sim \mathcal{N}(0, \sigma_n^2)$. The framework proceeds iteratively: (1) fit a probabilistic surrogate (typically a GP) to observed data $\mathcal{D}_t$; (2) select the next query point by maximizing an acquisition function $\alpha(\mathbf{x})$ that balances exploration and exploitation; (3) evaluate $f$ at the selected point and update $\mathcal{D}_{t+1}$. Common acquisition functions include Expected Improvement (EI) (Jones et al., 1998; Mockus, 1998), which quantifies expected gain over the current best $f^+ = \max_i y_i$: $\text{EI}(\mathbf{x}) = \mathbb{E}[\max(0, f(\mathbf{x}) - f^+)] = \sigma(\mathbf{x})[\gamma\Phi(\gamma) + \phi(\gamma)]$, where $\gamma = (\mu(\mathbf{x}) - f^+)/\sigma(\mathbf{x})$ and $\Phi, \phi$ denote the standard normal CDF and PDF. Upper Confidence Bound (UCB) (Srinivas et al., 2010; Auer et al., 2002) selects optimistically: $\text{UCB}(\mathbf{x}) = \mu(\mathbf{x}) + \beta_t^{1/2}\sigma(\mathbf{x})$, where $\beta_t$ is a theoretically-guided exploration parameter. Thompson Sampling (Thompson, 1933; Russo et al., 2018) draws a function $\tilde{f} \sim p(f \mid \mathcal{D}_t)$ from the posterior and optimizes it directly.

**Dirichlet Process and Mixture Models**   The Dirichlet Process (DP) (Ferguson, 1973; Antoniak, 1974) serves as the

cornerstone of Bayesian nonparametric modeling, providing a distribution over probability measures with support on an infinite sample space. It is rigorously defined via its finite-dimensional marginals.

**Definition 2.1** (Dirichlet Process). Let $(\Omega, \mathcal{F})$ be a measurable space and $G_0$ a probability measure on $\Omega$. A random probability measure $G$ is distributed according to a Dirichlet process with concentration parameter $\alpha > 0$ and base measure $G_0$, denoted $G \sim \mathrm{DP}(\alpha, G_0)$, if for every finite measurable partition $(A_1, \ldots, A_K)$ of $\Omega$, the vector of random probabilities follows a Dirichlet distribution: $\big(G(A_1), \ldots, G(A_K)\big) \sim \mathrm{Dir}\big(\alpha G_0(A_1), \ldots, \alpha G_0(A_K)\big)$. The parameter $\alpha$ governs the variance of the process; as $\alpha \to \infty$, $G$ converges weakly to $G_0$.

While Definition 2.1 establishes the existence of the process, it does not offer a direct method for sampling. The *stick-breaking construction* provides an explicit generative representation, proving that realizations of a DP are almost surely discrete.

**Theorem 2.2** (Sethuraman's Stick-Breaking Construction (Sethuraman, 1994)). *A random measure $G \sim \mathrm{DP}(\alpha, G_0)$ admits the almost sure representation:*

$$G = \sum_{k=1}^{\infty} \pi_k \delta_{\theta_k},$$

*where the atoms $\theta_k \overset{i.i.d.}{\sim} G_0$ and the weights $\{\pi_k\}$ are generated via:* $\beta_k \sim \mathrm{Beta}(1, \alpha); \pi_k = \beta_k \prod_{j=1}^{k-1}(1 - \beta_j)$.

This construction elucidates the clustering property of the DP: since $G$ is discrete, multiple observations $\theta_i \sim G$ will share identical values with non-zero probability. The number of unique values (clusters), denoted $K_n$, grows logarithmically with the dataset size $n$, allowing model complexity to adapt automatically to the data.

**Theorem 2.3** (Expected Number of Clusters). *For a sample of size $n$, the expected number of distinct clusters is*

$$\mathbb{E}[K_n \mid \alpha] = \sum_{i=1}^{n} \frac{\alpha}{i - 1 + \alpha},$$

*and admits the asymptotic expansion* $\mathbb{E}[K_n \mid \alpha] = \alpha \log(1 + n/\alpha) + O(1)$ *as* $n \to \infty$.

While the stick-breaking view describes the conditional distribution of observations given $G$, the *Chinese Restaurant Process* (CRP) describes the marginal distribution of cluster assignments obtained by integrating out $G$.

**Definition 2.4** (Chinese Restaurant Process). Given assignments $z_{1:n-1}$, for each existing cluster $k \in \{1, \ldots, K_{n-1}\}$,

$$p(z_n = k \mid z_{1:n-1}, \alpha) = \frac{n_k}{n - 1 + \alpha},$$

and the probability of creating a new cluster is

$$p(z_n = K_{n-1} + 1 \mid z_{1:n-1}, \alpha) = \frac{\alpha}{n - 1 + \alpha},$$

where $n_k = \#\{i < n : z_i = k\}$.

The CRP exhibits a rich-get-richer property (Pitman & Yor, 1997): popular clusters attract more members, inducing power-law cluster sizes. Combining these elements, a Dirichlet Process Mixture Model (DPMM) (Escobar & West, 1995; Neal, 2000; McLachlan & Peel, 2000) places a DP prior over mixture components: $G \sim \mathrm{DP}(\alpha, G_0)$, $\theta_i \sim G$, and $x_i \sim F(\theta_i)$ for some parametric family $F$. The resulting model has infinitely many potential components but instantiates only finitely many for any finite dataset.

## 3  Probabilistic Surrogate for RAMBO

### 3.1  Generative Process of DPMM

The Dirichlet Process Mixture of Gaussian Processes (DPMM-GP) models the objective as a countable mixture of independent GPs, partitioning the search space into latent "regimes" that adapt to non-stationarity and heteroscedasticity. Let $\alpha > 0$ denote the concentration parameter and $G_0(\theta)$ the base measure over kernel hyperparameters. The generative process first constructs mixture weights via stick-breaking: $\beta_k \sim \mathrm{Beta}(1, \alpha)$ with $\pi_k = \beta_k \prod_{j<k}(1 - \beta_j)$. Each regime $k$ is then assigned hyperparameters $\theta_k = \{\sigma_{f,k}^2, \ell_k, \sigma_{n,k}^2\} \sim G_0$. For each observation $i$, we draw a latent assignment $z_i \sim \mathrm{Categorical}(\{\pi_k\}_{k=1}^{\infty})$, then generate the observation from the corresponding GP: $f_k \sim \mathcal{GP}(0, k_{\theta_k})$ and $y_i \mid z_i = k \sim \mathcal{N}(f_k(\mathbf{x}_i), \sigma_{n,k}^2)$.

This hierarchical structure allows distinct components to capture qualitatively different local characteristics. The signal variance $\sigma_{f,k}^2$ governs the output amplitude, the length scale $\ell_k$ dictates the function's local smoothness, and $\sigma_{n,k}^2$ captures the local observation noise variance within regime $k$. The base distribution $G_0$ is factorized as a product of independent Inverse-Gamma priors (Gelman et al., 2013), which provide weakly-informative, positively-supported priors with finite moments: $\sigma_{f,k}^2 \sim \mathrm{InvGamma}(a_f, b_f); \ell_k \sim \mathrm{InvGamma}(a_\ell, b_\ell); \sigma_{n,k}^2 \sim \mathrm{InvGamma}(a_n, b_n)$. Note that while Inverse-Gamma is conjugate to a Gaussian variance in a direct observation model, it is not fully conjugate to the GP marginal likelihood $\mathcal{N}(\mathbf{0}, \mathbf{K}_k + \sigma_{n,k}^2 \mathbf{I})$. Posterior inference over $\theta_k$ is therefore carried out via gradient-based optimization (Adam) or Metropolis-Hastings rather than closed-form updates; the Inverse-Gamma family is chosen for its numerical stability and well-defined moments, not for analytical conjugacy. The hyperparameters for these priors are calibrated empirically based on the data range. We set the shape parameters $a_f = a_\ell = a_n = 2$ to ensure finite means, while the scale parameters $b_f, b_\ell, b_n$ are ad-

justed according to the empirical variance and input domain bounds.

**Joint Distribution and Marginal Likelihood**  We now formalize the generative structure and derive the marginal likelihood required for inference. The joint probability density of the DPMM-GP decomposes into the nonparametric prior over the mixture components and the conditional likelihood of the observations.

**Theorem 3.1** (DPMM-GP Joint Distribution). *Given the hyperparameters $\{\alpha, G_0\}$ and input data $\mathbf{X}$, the joint distribution over the latent variables and observations factorizes as:*

$$p(\mathbf{y}, \mathbf{z}, \Theta, \mathbf{f} \mid \mathbf{X}) = \left[ \prod_{k=1}^{\infty} p(\beta_k \mid \alpha) p(\theta_k \mid G_0) p(f_k \mid \theta_k) \right]$$
$$\times \prod_{i=1}^{n} \underbrace{p(z_i \mid \boldsymbol{\beta})}_{\pi_{z_i}} \underbrace{p(y_i \mid f_{z_i}(\mathbf{x}_i), \theta_{z_i})}_{\mathcal{N}(y_i \mid \dots)} \quad (1)$$

*where $\Theta = \{\theta_k, \beta_k\}_{k=1}^{\infty}$ represents the global parameters.*

A structural advantage of this model is the analytic tractability of the Gaussian Process. Since the GP prior is conjugate to the Gaussian likelihood, we can analytically integrate out the latent function values $\mathbf{f}$ to obtain a closed-form marginal likelihood for each cluster.

**Proposition 3.2** (Cluster Marginal Likelihood). *Let $\mathbf{y}_k$ and $\mathbf{X}_k$ denote the subset of observations assigned to regime $k$. The marginal likelihood for this cluster, conditioned on hyperparameters $\theta_k$, is:*

$$p(\mathbf{y}_k \mid \mathbf{X}_k, \theta_k) = \mathcal{N}(\mathbf{y}_k \mid \mathbf{0}, \mathbf{K}_k + \sigma_{n,k}^2 \mathbf{I}), \quad (2)$$

*where $[\mathbf{K}_k]_{ij} = k_{\theta_k}(\mathbf{x}_i, \mathbf{x}_j)$ is the kernel matrix evaluated on $\mathbf{X}_k$.*

*Proof.* The result follows from the standard convolution of Gaussians. The marginalization of the latent function $f_k$ is defined as:

$$\int \mathcal{N}(\mathbf{y}_k \mid \mathbf{f}_k, \sigma_{n,k}^2 \mathbf{I}) \mathcal{N}(\mathbf{f}_k \mid \mathbf{0}, \mathbf{K}_k) \, d\mathbf{f}_k. \quad (3)$$

Using standard Gaussian identities, the sum of independent Gaussian variables (signal $f_k$ plus noise $\varepsilon$) yields a Gaussian with covariance $\mathbf{K}_k + \sigma_{n,k}^2 \mathbf{I}$. $\square$

This closed-form marginalization reduces the inference problem to sampling only the discrete assignments $\mathbf{z}$ and hyperparameters $\Theta$, significantly improving MCMC mixing rates.

## 3.2 Posterior Inference via Collapsed Gibbs Sampling

Direct optimization of the DPMM-GP objective is intractable due to the discrete nature of the regime assignments and the trans-dimensional parameter space (the number of regimes $K$ is not fixed). Furthermore, standard Gibbs sampling schemes (Neal, 2000) that instantiate the latent function values $\mathbf{f}$ suffer from severe autocorrelation, as the strong coupling between $\mathbf{f}$ and $\mathbf{z}$ inhibits mixing (Liu, 1994; Neal, 2000; Ishwaran & James, 2001). By analytically marginalizing out $\mathbf{f}$ (Theorem 3.1), we reduce the state space to only the assignments $\mathbf{z}$ and hyperparameters $\Theta$. This "collapsed" scheme significantly improves mixing efficiency while retaining the ability to explore the multimodal posterior distribution of the partition structure.

**Sampling Regime Assignments**  For each observation $i$, we sample a new assignment $z_i$ conditioned on all other variables. The conditional probability of assigning observation $i$ to regime $k$ is proportional to the product of the CRP prior and the conditional likelihood:

$$p(z_i = k \mid \mathbf{z}_{-i}, \mathcal{D}) \propto$$
$$\underbrace{p(z_i = k \mid \mathbf{z}_{-i}, \alpha)}_{\text{CRP Prior}} \times \underbrace{p(y_i \mid \mathbf{x}_i, \mathcal{D}_{k,-i}, \theta_k)}_{\text{Likelihood}}. \quad (4)$$

Depending on the regime index $k$, this probability takes two forms:

1. **Existing Regime ($k \leq K$):** The prior probability is proportional to $n_{k,-i}$, the count of current members excluding $i$. The likelihood term is the GP posterior predictive density conditioned on the existing data in regime $k$:

$$p(y_i \mid \dots) = \mathcal{N}(y_i \mid \mu_{i|k}, \sigma_{i|k}^2), \quad (5)$$

where $\mu_{i|k}$ and $\sigma_{i|k}^2$ are the standard GP predictive mean and variance given $\mathcal{D}_{k,-i}$.

2. **New Regime ($k = K + 1$):** The prior probability is proportional to $\alpha$. The likelihood is the marginal probability under $G_0$. We approximate this via Monte Carlo integration:

$$p(y_i \mid \mathbf{x}_i, G_0) \approx \frac{1}{M} \sum_{m=1}^{M} \mathcal{N}(y_i \mid 0, \sigma_{tot}^{2(m)}), \quad (6)$$

where $\sigma_{tot}^{2(m)} = k_{\theta^{(m)}}(\mathbf{x}_i, \mathbf{x}_i) + \sigma_{n^{(m)}}^2$ represents the total variance for sample $\theta^{(m)} \sim G_0$.

**Updating Hyperparameters**  Given the assignments $\mathbf{z}$, the hyperparameters for each active regime are independent. We update $\theta_k$ by targeting the posterior $p(\theta_k \mid \mathcal{D}_k) \propto p(\mathbf{y}_k \mid \theta_k) p(\theta_k \mid G_0)$. In our experiments, we adopt an empirical Bayes approach, maximizing the log-marginal likelihood

w.r.t. $\theta_k$ using Adam (Kingma & Ba, 2015) for computational efficiency. Alternatively, a fully Bayesian treatment can be achieved via Metropolis-Hastings (Hastings, 1970) steps with log-normal proposals, accepting updates based on the marginal likelihood ratio. We defer the complete inference procedure to Algorithm 1 in the Appendix C.1.

### 3.3 Posterior Predictive Distribution

Given a posterior sample of regime assignments $\mathbf{z}$ and hyperparameters $\Theta$ from the collapsed Gibbs sampler, the predictive distribution at a new test point $\mathbf{x}_*$ is obtained by marginalizing over the latent assignment $z_*$. We first state the exact predictive that follows directly from the generative model, then introduce a predictive modeling choice that restores spatial relevance for multi-regime landscapes.

**Theorem 3.3** (Exact DPMM-GP Posterior Predictive). *Conditional on a posterior sample $(\mathbf{z}, \Theta)$, the predictive density for a test input $\mathbf{x}_*$ marginalized over its latent assignment is*

$$p(y_* \mid \mathbf{x}_*, \mathbf{z}, \Theta, \mathcal{D}) = \sum_{k=1}^{K+1} \pi_k \cdot \mathcal{N}(y_* \mid \mu_{*,k}, \sigma_{*,k}^2), \quad (7)$$

*where the predictive weights are the underlined input-independent CRP marginals*

$$\pi_k = \tfrac{n_k}{n+\alpha} \ (k \le K), \qquad \pi_{K+1} = \tfrac{\alpha}{n+\alpha}, \quad (8)$$

*and $(\mu_{*,k}, \sigma_{*,k}^2)$ are the standard GP posterior mean and variance under regime $k$; for $k = K + 1$, the component is the prior predictive under $G_0$, approximated by Monte Carlo. The full derivation is provided in Appendix A.*

**Why a Spatial Modulation Is Needed.** Theorem 3.3 aggregates regimes by their global popularity $\pi_k$, independent of where $\mathbf{x}_*$ falls. For multi-regime scientific landscapes—where each regime is locally coherent but spatially confined (e.g., distinct molecular scaffolds, magnetic topologies, or torsional basins)—this aggregation averages over regimes that are irrelevant at $\mathbf{x}_*$, producing under-confident predictions whenever regimes occupy disjoint spatial supports. The operational question at prediction time is which regime is relevant at this test point, and how much should it contribute?—a distinction the CRP marginals alone cannot make. We therefore adopt the following predictive modeling choice:

**Proposition 3.4** (Spatially-Modulated Predictive Weights). *Replacing the CRP weights $\pi_k$ in Eq. (7) with*

$$w_k(\mathbf{x}_*) \ \propto \ \pi_k \cdot \sigma_{*,k}^{-1}(\mathbf{x}_*) \ = \ \frac{n_k}{n + \alpha} \cdot \exp\left(-\tfrac{1}{2} \log \sigma_{*,k}^2(\mathbf{x}_*)\right) \quad (9)$$

*yields a predictive distribution that admits two independent justifications: (i) it is proportional to the expected posterior responsibility $\mathbb{E}_{y_*}\left[n_k \cdot \mathcal{N}(y_* \mid \mu_{*,k}, \sigma_{*,k}^2)\right]$ evaluated under each component's own predictive at $\mathbf{x}_*$; and*

*(ii) it corresponds to a Jeffreys' (scale-invariant) reference measure for aggregation across components with heterogeneous predictive scales $\sigma_{*,k}$. The construction introduces no additional learnable parameters—the spatial dependence arises directly from the GP posterior scales. The derivation appears in Appendix A.*

The first factor $\frac{n_k}{n+\alpha}$ retains the "rich-get-richer" property of the Dirichlet Process, while $\sigma_{*,k}^{-1}(\mathbf{x}_*)$ acts as a spatial confidence weight, suppressing regimes that are uncertain at $\mathbf{x}_*$. Unlike input-dependent gating networks (Rasmussen & Ghahramani, 2001), Eq. (9) introduces no new parameters: the spatial modulation is induced by the GP posterior itself. We use these weights $w_k(\mathbf{x})$ in all downstream predictive and acquisition computations (Sections 4 and B).

**Theorem 3.5** (Moment Matching). *The mean $\mu_{mix}$ and variance $\sigma_{mix}^2$ of the predictive mixture are:*

$$\mu_{mix}(\mathbf{x}_*) = \sum_{k=1}^{K+1} w_k(\mathbf{x}_*)\mu_{*,k}, \quad (10)$$

$$\sigma_{mix}^2(\mathbf{x}_*) = \sum_{k=1}^{K+1} w_k(\mathbf{x}_*)\left[\sigma_{*,k}^2 + \mu_{*,k}^2\right] - \mu_{mix}^2. \quad (11)$$

*Proof.* We apply the laws of total expectation and variance. Let $Z$ be the indicator variable for the regime assignment.

1. $\mathbb{E}[y_*] = \mathbb{E}_Z[\mathbb{E}[y_* \mid Z]] = \sum_k w_k \mu_{*,k}$.

2. $\mathbb{V}\mathrm{ar}[y_*] = \mathbb{E}_Z[\mathbb{V}\mathrm{ar}[y_* \mid Z]] + \mathbb{V}\mathrm{ar}_Z[\mathbb{E}[y_* \mid Z]]$.

Substituting the component moments:

$$\sigma_{\mathrm{mix}}^2 = \sum_k w_k \sigma_{*,k}^2 + \left(\sum_k w_k \mu_{*,k}^2 - \left(\sum_k w_k \mu_{*,k}\right)^2\right)$$
$$= \sum_k w_k(\sigma_{*,k}^2 + \mu_{*,k}^2) - \mu_{\mathrm{mix}}^2. \quad (12)$$
$\square$

This variance decomposition highlights two distinct sources of uncertainty. The first term, $\sum w_k \sigma_{*,k}^2$, represents the **intra-regime uncertainty** (average GP variance). The second term, $\mathbb{V}\mathrm{ar}_Z[\mu]$, captures the **inter-regime disagreement** (variance of the means). This ensures robust uncertainty quantification: even if individual regimes are confident, the model reports high overall uncertainty if the regimes disagree on the prediction.

## 4 Acquisition Functions

To guide the optimization process, we must map the posterior predictive distribution to a scalar utility value. We

prioritize Expected Improvement (EI) due to its analytic tractability and robustness to the non-stationary scaling inherent in our mixture model.

**Mixture Expected Improvement**   In the DPMM-GP framework, different latent regimes often exhibit vastly different signal variances ($\sigma_{f,k}^2$). A regime modeling a "flat" region may have small variance, while a "rough" regime has large variance. Metric-based acquisition functions like UCB require a trade-off parameter $\beta_t$ that is difficult to calibrate across these heterogeneous scales.

In contrast, Expected Improvement naturally adapts to local scaling. It quantifies the expected gain over the current best observation $f^+ = \max_i y_i$, weighted by the probability of the latent regime assignment.

**Theorem 4.1** (DPMM-GP Expected Improvement). *Using the spatially-modulated predictive weights $w_k(\mathbf{x})$ of Proposition 3.4 (Eq. (9)), the Expected Improvement at input $\mathbf{x}$ is the weight-summed EI of each constituent GP component:*

$$\alpha_{EI}(\mathbf{x}) = \sum_{k=1}^{K+1} w_k(\mathbf{x}) \cdot \sigma_{*,k}(\mathbf{x}) \left[ \gamma_k \Phi(\gamma_k) + \phi(\gamma_k) \right], \quad (13)$$

*where $\gamma_k = (\mu_{*,k}(\mathbf{x}) - f^+)/\sigma_{*,k}(\mathbf{x})$ is the normalized improvement Z-score for regime $k$, and $\Phi, \phi$ are the standard normal CDF and PDF, respectively.*

*Proof.* Let $I(\mathbf{x}) = \max(0, f(\mathbf{x}) - f^+)$ be the improvement utility. Treating the modulated weights $w_k(\mathbf{x})$ of Eq. (9) as the predictive probability assigned to $z_* = k$ at $\mathbf{x}$ and applying the Law of Total Expectation,

$$\mathbb{E}[I(\mathbf{x})] = \mathbb{E}_{z_*}[\mathbb{E}[I(\mathbf{x}) \mid z_*]] = \sum_{k=1}^{K+1} w_k(\mathbf{x}) \cdot \mathbb{E}_{\text{GP}_k}[I(\mathbf{x})].$$
$$(14)$$

Since each conditional component is Gaussian, $\mathbb{E}_{\text{GP}_k}[I(\mathbf{x})]$ reduces to the standard closed-form GP-EI formula. We emphasize that the weights here are the modulated $w_k(\mathbf{x})$ from Eq. (9), not the input-independent CRP marginals $\pi_k$ in Eq. (8). $\qquad \square$

This formulation encourages a balanced search strategy: the acquisition value is high if a point belongs to a regime that predicts high improvement, or if there is significant ambiguity about the regime assignment itself (via weights $w_k$), necessitating exploration to resolve the structural uncertainty.

**Adaptive Concentration Parameter Scheduling**   The concentration parameter $\alpha$ governs regime creation: larger $\alpha$ encourages more clusters, with $\mathbb{E}[K_n \mid \alpha] \approx \alpha \log(n/\alpha + 1)$ (Antoniak, 1974). Prior work either fixes $\alpha$ or learns it

via MCMC (Rasmussen & Ghahramani, 2001), but for sequential optimization, the appropriate $\alpha$ varies with data availability—early stages benefit from small $\alpha$ to avoid premature fragmentation when observations are sparse, while later stages permit larger $\alpha$ to discover fine-grained structure as evidence accumulates. We formalize this intuition by deriving a schedule that matches the prior's expected complexity to a polynomial regime discovery rate $\mathcal{O}(n^\beta)$. Setting $\beta = 1/2$, motivated by square-root growth laws observed in clustering and information retrieval (Heaps, 1978), yields the **Log-Sqrt Schedule**:

$$\alpha_t = \alpha_0 \cdot \frac{\sqrt{t}}{\log(t + e)}, \quad (15)$$

where $\alpha_0 = 0.2$ is the base concentration. This schedule enforces early parsimony, enables progressive refinement, and balances bias-variance throughout optimization. The full derivation is provided in Appendix F.

**Acquisition Optimization**   The acquisition landscape of the DPMM-GP is inherently multimodal, inheriting local optima from the superposition of multiple regime-specific GP posteriors. Consequently, standard convex optimization is insufficient. We maximize $\alpha(\mathbf{x})$ using a multi-start L-BFGS-B strategy (Wilson et al., 2018) with automatic differentiation. To ensure robust convergence, the optimizer is initialized with a hybrid set of candidates $\mathcal{S}_{\text{init}}$ comprising: (1) a dense set of uniform random samples from $\mathcal{X}$ to encourage global exploration, (2) the centroids of currently active regimes to exploit high-probability regions, and (3) local Gaussian perturbations around the current best observation $\mathbf{x}^*$. The complete optimization loop is detailed in Algorithm 2 (Appendix C.2). While we prioritize EI for its parameter-free formulation and natural robustness to the heterogeneous scales, the closed-form mixture moments derived in Theorem 3.5 enable seamless extension to other standard acquisition functions—including UCB, Thompson Sampling, Max-value Entropy Search (Wang & Jegelka, 2017), Knowledge Gradient (KG) (Frazier et al., 2009), Predictive Entropy Search (PES) and Probability of Improvement (Kushner, 1964)—without additional approximation (see Appendix B for details).

## 5   Related Work

Modern Bayesian optimization (BO) is founded on Expected Improvement (Jones et al., 1998) and the theoretical regret analysis for GP bandit optimization—comprising the original GP-UCB upper bound (Srinivas et al., 2010), improved analyses with kernel-dependent rates (Chowdhury & Gopalan, 2017), and algorithm-independent lower bounds (Scarlett et al., 2017)—with practical adoption driven by efficient GP implementations (Snoek et al., 2012). Standard acquisition strategies have expanded to include

Thompson Sampling (Thompson, 1933; Russo et al., 2018), Knowledge Gradient (Frazier et al., 2009; Wu & Frazier, 2016), and entropy-based search (Hennig & Schuler, 2012; Wang & Jegelka, 2017; Hernández-Lobato et al., 2014). While robust, these methods typically assume stationary surrogates. To address scalability and high-dimensionality, recent approaches employ local trust regions (TuRBO (Eriksson et al., 2019), SCBO (Eriksson & Poloczek, 2021)), dimensionality reduction via embeddings (REMBO (Wang et al., 2016), ALEBO (Letham et al., 2020), BAxUS (Papenmeier et al., 2022)), or structural priors (SAASBO (Eriksson & Jankowiak, 2021), Add-GP-UCB (Kandasamy et al., 2015)). Although recent studies (Xu et al., 2025; Hvarfner et al., 2024) suggest standard GPs with proper initialization can rival these specialized methods, they do not resolve the limitations of stationarity in heterogeneous landscapes. Multi-fidelity methods (Poloczek et al., 2017; Takeno et al., 2020; Huang et al., 2006; Kandasamy et al., 2016; Li et al., 2020; 2021) exploit cheap approximations to accelerate optimization, though they assume consistent structure across fidelity levels. Parallel advances extend BO to mixed-variable and combinatorial spaces using random forests (Hutter et al., 2011; Bergstra et al., 2011), tree-structured estimators (Bergstra et al., 2011; Falkner et al., 2018), and graph kernels (Oh et al., 2019; Wan et al., 2021; Garrido-Merchán & Hernández-Lobato, 2020; Deshwal & Doppa, 2021; Ru et al., 2020). Specific remedies for non-stationarity include input warping (HEBO (Cowen-Rivers et al., 2022), Warped GPs (Snoek et al., 2014)), Deep GPs (Damianou & Lawrence, 2013; Wilson et al., 2016b), input-dependent kernels (Paciorek & Schervish, 2003), neural surrogates (Snoek et al., 2015; Springenberg et al., 2016; White et al., 2021), and axis-aligned partitioning via Treed GPs (Gramacy & Lee, 2008). However, these approaches generally rely on single global models or hard geometric partitions, failing to quantify the probabilistic uncertainty inherent in natural regime boundaries. *Most related* to our work is the Infinite Mixture of GP Experts (Rasmussen & Ghahramani, 2001), with other mixture-of-GP formulations explored in (Tresp, 2000b;a; Meeds & Osindero, 2005; Yuan & Neubauer, 2008; Nguyen & Bonilla, 2014; Gadd et al., 2020; Li & Ma, 2023). The critical distinction is in the gating mechanism: Rasmussen & Ghahramani employ an *input-dependent* gating network that conditions regime assignment on spatial location, requiring explicit learning of gating boundaries and introducing additional gating parameters; enriched variants (Gadd et al., 2020) likewise modulate the gate via auxiliary covariates. Our framework instead uses an *input-independent* Chinese Restaurant Process prior in the generative model (Theorem 3.3) and introduces spatial dependence only at prediction time as a predictive modeling choice (Proposition 3.4): the spatial modulation $\sigma_{*,k}^{-1}(\mathbf{x}_*)$ is read off directly from each regime's GP posterior, introducing no new learnable parameters and no gradient flow into a gating

network. Closer to ours, Li & Ma (2023) apply a DPMM of GPs to supervised functional regression with variational EM; in contrast, RAMBO targets sequential black-box optimization—the surrogate must support repeated, low-data refits and feed acquisition functions, motivating our collapsed Gibbs sampler with analytically marginalized latent functions, adaptive $\alpha$-scheduling, regime-aware acquisition (Theorem 4.1), and regime-centroid initialization for acquisition optimization—none of which arise in the supervised regression setting. RAMBO thus adapts this non-parametric philosophy specifically for Bayesian Optimization through regime-aware acquisition functions—decomposing uncertainty into intra-regime aleatoric variance and inter-regime epistemic disagreement—and adaptive $\alpha$-scheduling that enforces parsimony early while enabling fine-grained regime discovery as data accumulates.

# 6  Experiments

We evaluate RAMBO against state-of-the-art Bayesian optimization baselines across tasks spanning synthetic functions to complex scientific applications in structural chemistry, molecular biology, and nuclear fusion, specifically selected to assess surrogate model performance under conditions of high dimensionality, severe multi-modality, and heterogeneous landscapes—scenarios where standard stationary GPs typically flounder.

**Competing Methods**   We compare RAMBO against a diverse set of state-of-the-art algorithms, beginning with *Standard Single-GP BO (SGP)* (Snoek et al., 2012); to ensure a fair comparison and isolate the impact of our regime-adaptive mechanism, we employ the standard Squared Exponential kernel for both the SGP baseline and RAMBO. To assess performance in high-dimensional and heterogeneous settings, we include *TuRBO*[1] (Eriksson et al., 2019), which restricts optimization to local trust regions to handle non-stationarity; *SAASBO*[2] (Eriksson & Jankowiak, 2021), which addresses high dimensionality via sparse axis-aligned subspace priors; *BAxUS*[3] (Papenmeier et al., 2022), which progressively expands the dimensionality of its adaptive subspace embeddings; and *ALEBO*[4] (Letham et al., 2020), which optimizes within a linear embedding of the input space. Our evaluation also incorporates *HEBO*[5] (Cowen-Rivers et al., 2022), a robust method combining heteroscedastic GPs with input warping; *Bounce*[6] (Papenmeier et al., 2023), a trust-region approach designed for high-

---

[1] https://github.com/uber-research/TuRBO

[2] https://github.com/martinjankowiak/saasbo

[3] https://github.com/lpapenme/BAxUS

[4] https://github.com/facebookresearch/alebo

[5] https://github.com/huawei-noah/HEBO

[6] https://github.com/lpapenme/bounce

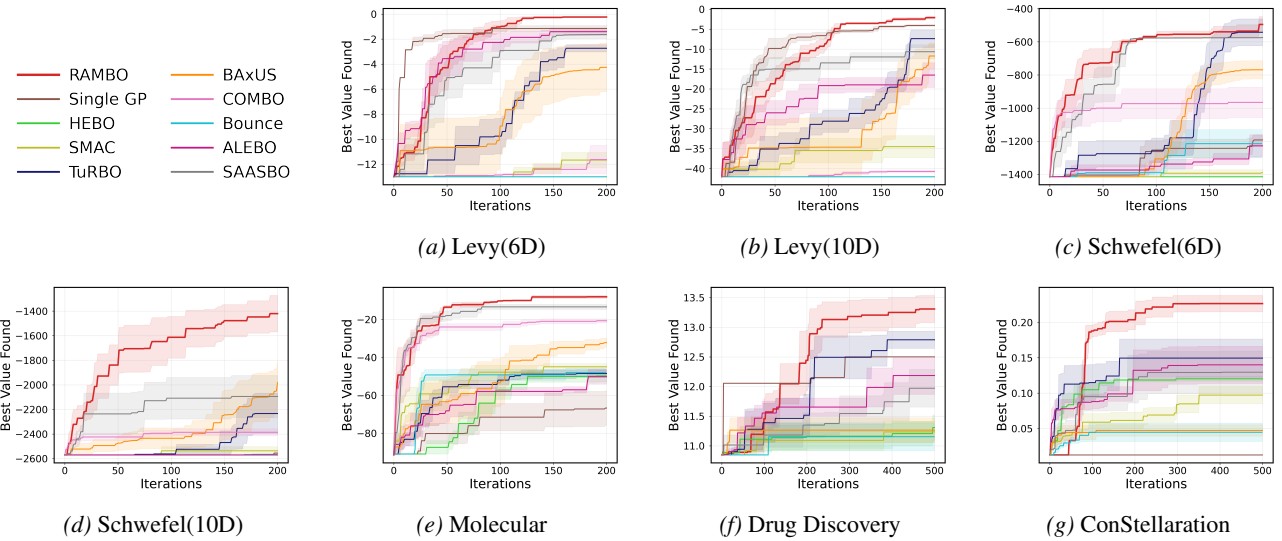

*Figure 1.* Optimization performance across synthetic and real-world benchmarks. We report the best objective value found (mean $\pm$ SE over 5 seeds). **(a)–(d)** Levy and Schwefel functions in 6D and 10D. **(e)–(g)** Molecular conformer optimization (12D), virtual screening for drug discovery (50D), and stellarator reactor design (80D). RAMBO consistently matches or outperforms all baselines, with the largest gains on *high-dimensional*, *multi-regime* landscapes.

dimensional mixed spaces; *SMAC*[7] (Hutter et al., 2011), a random-forest-based alternative for non-smooth landscapes; and *COMBO*[8] (Oh et al., 2019), which utilizes graph kernels to model combinatorial variables. Note that COMBO is omitted from the Drug Discovery and ConStellaration experiments because its computational complexity renders it intractable on these large-scale benchmarks.

**Metrics and Experiment Settings**  We report the best objective value found so far averaged over *5* independent seeds, with shaded regions denoting the *standard error*. All methods are initialized with (20 for synthetic and 5 for scientific design) identical quasirandom Sobol points. To ensure numerical stability, input features are normalized to the hypercube $[-1, 1]^d$, and target values are standardized (zero mean, unit variance) prior to model fitting in each iteration. RAMBO and the SGP baseline are implemented in PyTorch (Paszke et al., 2019), utilizing the *Expected Improvement (EI)* acquisition function. We optimize the acquisition function using L-BFGS with *20 random restarts*, selecting the candidate with the highest acquisition value to update the dataset. For RAMBO, cluster-specific kernel parameters are optimized via Adam (learning rate 0.05, 200 gradient steps per BO iteration, no early stopping) maximizing the marginal log-likelihood; inference is performed via collapsed Gibbs sampling with 500 burn-in iterations followed by $S = 5$ post-burn-in posterior samples used for the acquisition computation. The pruning threshold is

$\epsilon = 10^{-3}$. Other baselines use their official implementations; all experiments except SAASBO ran on *CPU-only* resources.

### 6.1  Synthetic Benchmarks

We validate RAMBO on two canonical test functions designed to stress-test optimization in pathological multimodal landscapes. The *Levy function*[9] is characterized by a rugged surface with dense clusters of local minima, challenging the model's ability to resolve fine-grained structures and navigate high-frequency oscillations. In contrast, the *Schwefel function*[10] presents a deceptive topology where the global optimum is geometrically isolated at the domain boundary, far removed from other local basins; this penalizes methods that over-exploit central regions. Formal definitions and visualizations are provided in Appendix D.1 and D.2. We evaluate performance across varying dimensions $d \in \{6, 10\}$ to assess scalability across distinct regimes: from low-dimensional settings that allow for visual verification of regime discovery, to high-dimensional spaces where standard stationary GPs typically falter. As shown in Figure 1a-1d, RAMBO with adaptive $\alpha$-scheduling (DPMM-Sched) consistently matches or outperforms all baselines across both functions and dimensionalities. On the Levy function, RAMBO converges to near-optimal values significantly faster than competing methods, while on the deceptive Schwefel landscape, the performance gap widens substantially—particularly in 10D, where stationary methods

[7]https://github.com/automl/SMAC3
[8]https://github.com/QUVA-Lab/COMBO
[9]https://www.sfu.ca/~ssurjano/levy.html
[10]https://www.sfu.ca/~ssurjano/schwef.html

struggle to escape suboptimal basins. Notably, the scheduled $\alpha$ variant outperforms fixed-$\alpha$ configurations, validating the benefit of adaptive regime discovery.

## 6.2 Real-World Scientific Design

**Molecular Conformer Optimization (12D)** Molecules continuously explore conformational space through rotations around single bonds (Hawkins et al., 2010; Riniker & Landrum, 2015). This benchmark involves finding the lowest-energy configuration of a linear alkane chain (pentadecane, $C_{15}H_{32}$) by optimizing $d = 12$ internal dihedral angles using force field calculations (Halgren, 1996; Grimme et al., 2017). Rotation around each C–C bond favors three orientations—$180°$ (anti) and $\pm 60°$ (gauche)—inducing a combinatorial explosion of locally stable states. With $3^{12} = 531,441$ potential conformational minima separated by high-energy steric barriers, the landscape is highly multimodal with sharp transitions between basins. A detailed problem description is provided in Appendix D.3. Figure 1e demonstrates that RAMBO substantially outperforms all baselines on this 12D conformational landscape. After 200 iterations, RAMBO achieves an energy 8.05 kcal/mol, compared to 13.36 kcal/mol for the best baseline (SAASBO), representing a 39.73% improvement. Most competing methods stagnate at high-energy local minima, whereas RAMBO —particularly with adaptive $\alpha$-scheduling—efficiently navigates between distinct rotameric basins to reach near-optimal configurations within 50 iterations.

**Virtual Screening for Drug Discovery (50D)** We optimize small molecules for docking scores against a cancer-related protein target (PDB ID: 6T2W). The inputs are 2048-bit Morgan fingerprints (Rogers & Hahn, 2010) compressed into a $d = 50$ dimensional continuous latent space via Principal Component Analysis (PCA). This benchmark is representative of modern drug discovery pipelines (Gómez-Bombarelli et al., 2018; Griffiths & Hernández-Lobato, 2020; Korovina et al., 2020; Stanton et al., 2022; Liu et al., 2023; 2025). This benchmark poses two key challenges: high dimensionality and the inherently disjoint nature of chemical space—distinct molecular scaffolds (e.g., different ring systems or functional groups) occupy separate regions in the latent space with fundamentally different structure-activity relationships (SAR). A detailed problem description is provided in Appendix D.4. As shown in Figure 1f, RAMBO substantially outperforms baselines in this 50D drug discovery task. By iteration 500, RAMBO achieves a docking score of -13.31, compared to -12.79 for the next-best method (TuRBO), representing a 4.06% improvement in predicted binding affinity. The mixture model's ability to partition chemical space into scaffold-specific regimes enables continued improvement throughout the 500-iteration

budget, while single-surrogate methods stagnate in early iterations.

**Nuclear Fusion Reactor (80D)** This benchmark optimizes the shape of a stellarator fusion reactor to maximize quasi-isodynamic quality, a measure of particle confinement efficiency (Gates et al., 2018; Cadena et al., 2025). The inputs are Fourier coefficients controlling the plasma boundary geometry. Stellarator design exhibits highly nonlinear physics: small perturbations in boundary shape can trigger abrupt transitions in magnetic field topology and plasma stability, creating a patchy landscape where regions of high confinement quality are interspersed with unstable configurations. A detailed problem description is provided in Appendix D.5. Figure 1g shows that RAMBO achieves substantially higher confinement quality than all baselines, reaching $Q_i$ values approximately 51.55% higher than the best competing method. RAMBO discovers 3-5 regimes corresponding to distinct magnetic topologies, modeling configurations near nested flux surfaces separately from those approaching island formation. Stable high-$Q_i$ regions receive moderate length scales, while boundary regions near instabilities are modeled with shorter length scales to capture rapid quality degradation. Embedding-based methods (ALEBO, SAASBO) struggle because stellarator physics involves dense interactions among Fourier coefficients, violating both the low-rank linear structure assumed by ALEBO and the axis-aligned sparsity assumed by SAASBO. Trust-region methods (TuRBO) plateau when the local region straddles a stability boundary. RAMBO's regime-aware surrogates avoid these failure modes, enabling continued improvement throughout the 500-iteration budget.

## 7 Conclusion

We presented RAMBO, a regime-adaptive Bayesian optimization framework replacing the stationary GP surrogate with a Dirichlet Process Mixture of Gaussian Processes. By discovering distinct regimes with locally-optimized hyperparameters, RAMBO captures discrete heterogeneity in scientific design problems where non-stationary kernels fall short. Collapsed Gibbs sampling enables efficient inference, while adaptive $\alpha$-scheduling balances parsimony against expressiveness. Experiments on synthetic and real-world benchmarks—molecular conformation, drug discovery, and fusion reactor design—demonstrate consistent improvements over state-of-the-art baselines.

## Acknowledgements

This work was supported by startup funding from the Department of Computer Science at Florida State University.

## Impact Statement

RAMBO is intended to accelerate scientific discovery in domains where each function evaluation is expensive—molecular conformer search, virtual screening for drug discovery, and stellarator fusion reactor design—by reducing the number of costly simulations or experiments required to reach high-quality solutions. The broader positive impact includes lowering the resource and energy footprint of computational science workflows, and broadening access to model-based design for groups without large compute budgets.

Potential risks warrant consideration. First, like all Bayesian optimization methods, RAMBO produces point recommendations under uncertainty; if downstream decision-makers treat acquisition values as ground-truth rankings rather than as one input to a deliberative process, decisions in safety-critical applications (e.g., drug candidate prioritization) can be miscalibrated. We mitigate this by explicitly decomposing predictive uncertainty into intra-regime variance and inter-regime disagreement (Section 3.3), making the uncertainty structure interpretable rather than collapsing it into a single scalar. Second, accelerated optimization in drug discovery or materials design can be applied to harmful as well as beneficial targets; we encourage downstream practitioners to follow domain-specific safety review processes when applying the method to novel design problems. Third, while the benchmarks we evaluate are public, real-world deployment may interact with proprietary datasets; we recommend audit of training data provenance prior to use in any consequential decision pipeline. The method does not involve human subjects, personally identifiable data, or generative models that synthesize text or images.

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

# A  Proof of Theorem 3.3 and Derivation of Proposition 3.4

We provide the complete derivation in two parts: (A) the exact DPMM-GP posterior predictive of Theorem 3.3 using the input-independent CRP marginals; and (B) the derivation of the spatially-modulated weights of Proposition 3.4 as a predictive modeling choice motivated by two independent and consistent perspectives—expected posterior responsibility and Jeffreys' scale-invariant aggregation. We emphasize that Part B is a deliberate structural augmentation of the predictive step that targets spatial relevance; the underlying generative DP prior remains input-independent throughout.

## A.1  Part A: Exact DPMM-GP Predictive (Theorem 3.3)

Given training data $\mathcal{D}$, a posterior sample of regime assignments $\mathbf{z}$, and hyperparameters $\Theta$ from the collapsed Gibbs sampler, we derive the predictive density at a test input $\mathbf{x}_*$ by marginalizing its latent assignment $z_*$ over the $K + 1$ possible regimes (the $K$ active regimes plus a new one).

**Step 1 (Law of total probability).**

$$p(y_* \mid \mathbf{x}_*, \mathbf{z}, \Theta, \mathcal{D}) = \sum_{k=1}^{K+1} p(z_* = k \mid \mathbf{z}, \alpha) \cdot p(y_* \mid \mathbf{x}_*, z_* = k, \mathcal{D}, \Theta). \tag{16}$$

**Step 2 (CRP predictive marginals).**  By Definition 2.4, the assignment of $z_*$ depends only on the cluster sizes $\{n_k\}$ in $\mathbf{z}$:

$$p(z_* = k \mid \mathbf{z}, \alpha) = \frac{n_k}{n + \alpha} \ \ (k \leq K), \qquad p(z_* = K + 1 \mid \mathbf{z}, \alpha) = \frac{\alpha}{n + \alpha}. \tag{17}$$

These probabilities are input-independent—they do not depend on $\mathbf{x}_*$.

**Step 3 (GP posterior predictive within each regime).**  Since the regime-conditional latent function $f_k$ has been analytically marginalized (Proposition 3.2), the per-regime predictive for $k \leq K$ is the standard GP posterior on regime $k$'s data $\mathcal{D}_k$ with hyperparameters $\theta_k$:

$$p(y_* \mid \mathbf{x}_*, z_* = k, \mathcal{D}, \Theta) = \mathcal{N}(y_* \mid \mu_{*,k}(\mathbf{x}_*), \sigma_{*,k}^2(\mathbf{x}_*)). \tag{18}$$

For $k = K + 1$, the predictive is the marginal under the base measure $G_0$, approximated via Monte Carlo over $G_0$-samples as in Section 3.2.

**Step 4 (Substitution).**  Combining Steps 2 and 3,

$$p(y_* \mid \mathbf{x}_*, \mathbf{z}, \Theta, \mathcal{D}) = \sum_{k=1}^{K} \frac{n_k}{n + \alpha} \mathcal{N}(y_* \mid \mu_{*,k}, \sigma_{*,k}^2) + \frac{\alpha}{n + \alpha} p(y_* \mid \mathbf{x}_*, \text{new}), \tag{19}$$

which is exactly Eq. (7) with the input-independent CRP weights $\pi_k$ of Eq. (8). □

## A.2  Part B: Derivation of the Spatially-Modulated Weights (Proposition 3.4)

Theorem 3.3 weights every regime by its global popularity $\pi_k = n_k/(n + \alpha)$, independent of $\mathbf{x}_*$. For multi-regime landscapes where each regime is spatially confined, this aggregation forces every regime to contribute even when most are irrelevant at $\mathbf{x}_*$. We construct the spatially-modulated weight $w_k(\mathbf{x}_*)$ of Eq. (9) as a predictive modeling choice motivated by two independent perspectives that produce the same functional form.

**Perspective 1: Expected Posterior Responsibility.**  If a test observation $y_*$ were available, the standard mixture-of-Gaussians responsibility (analogous to Eq. (4)) would assign

$$p(z_* = k \mid y_*, \mathbf{x}_*, \mathbf{z}, \Theta, \mathcal{D}) \propto n_k \cdot \mathcal{N}(y_* \mid \mu_{*,k}, \sigma_{*,k}^2). \tag{20}$$

Since $y_*$ is unobserved, we evaluate the expected unnormalized responsibility under component $k$'s own predictive at $\mathbf{x}_*$:

$$\mathbb{E}_{y_* \sim \mathcal{N}(\mu_{*,k}, \sigma_{*,k}^2)} \left[ n_k \cdot \mathcal{N}(y_* \mid \mu_{*,k}, \sigma_{*,k}^2) \right] = n_k \int \mathcal{N}(y \mid \mu_{*,k}, \sigma_{*,k}^2)^2 \, dy. \tag{21}$$

Using the Gaussian self-evaluation identity $\int \mathcal{N}(y \mid a, A)\,\mathcal{N}(y \mid b, B)\,dy = \mathcal{N}(a \mid b, A + B)$ with $a = b = \mu_{*,k}$, $A = B = \sigma^2_{*,k}$,

$$\int \mathcal{N}(y \mid \mu_{*,k}, \sigma^2_{*,k})^2 \, dy = \mathcal{N}(0 \mid 0, 2\sigma^2_{*,k}) = \frac{1}{2\sqrt{\pi}\,\sigma_{*,k}}. \tag{22}$$

Hence the expected unnormalized responsibility scales as $n_k \cdot \sigma^{-1}_{*,k}(\mathbf{x}_*)$, and incorporating the CRP normalization $1/(n + \alpha)$ yields

$$w_k(\mathbf{x}_*) \;\propto\; \frac{n_k}{n + \alpha} \cdot \sigma^{-1}_{*,k}(\mathbf{x}_*), \tag{23}$$

which is Eq. (9) via $\sigma^{-1}_{*,k} = \exp(-\tfrac{1}{2}\log \sigma^2_{*,k})$.

**Cross-regime terms.** A fully Bayesian responsibility would include cross-regime contributions $\mathbb{E}_{y_* \sim \mathcal{N}(\mu_{*,j}, \sigma^2_{*,j})}[\mathcal{N}(y_* \mid \mu_{*,k}, \sigma^2_{*,k})] = \mathcal{N}(\mu_{*,j} \mid \mu_{*,k}, \sigma^2_{*,j} + \sigma^2_{*,k})$ for $j \neq k$. These are exponentially suppressed whenever well-separated regimes satisfy $|\mu_{*,j} - \mu_{*,k}|^2 \gg \sigma^2_{*,j} + \sigma^2_{*,k}$, which is precisely the regime of interest (sharply heterogeneous landscapes); neglecting them preserves the leading-order behavior. The construction is thus exact in the well-separated limit and a controlled approximation otherwise.

**Perspective 2: Jeffreys' Scale-Invariant Reference Measure.** The aggregation in Eq. (7) mixes Gaussian components with heterogeneous predictive scales $\sigma_{*,k}$. The natural uninformative reference measure for a positive scale parameter is Jeffreys' prior $p_J(\sigma) \propto \sigma^{-1}$ (Jeffreys, 1946), which is invariant under reparameterizations of $\sigma$. Using $\sigma^{-1}_{*,k}(\mathbf{x}_*)$ as the per-component reference weight at $\mathbf{x}_*$, so that locally confident regimes (small $\sigma_{*,k}$) are up-weighted and locally uncertain ones contribute less. Combined with the CRP popularity $n_k/(n + \alpha)$, this perspective independently arrives at the same Eq. (9).

**No Additional Parameters.** Unlike input-dependent gating networks (Rasmussen & Ghahramani, 2001), which require learning gating boundaries via auxiliary parameters or networks, $w_k(\mathbf{x}_*)$ is constructed entirely from quantities already produced by the regime-conditional GP posteriors $(\mu_{*,k}, \sigma_{*,k})$. No new parameters, no auxiliary optimization, and no gradient flow into the gating mechanism are introduced. $\qquad\square$

# B  Extended Acquisition Functions

In this section, we provide the detailed derivations for the Max-value Entropy Search (MES) and Probability of Improvement (PI) acquisition functions within the RAMBO framework. Both derivations address the challenge of the multimodal posterior predictive distribution inherent to the DPMM-GP.

## B.1  Max-value Entropy Search (MES)

We extend the Max-value Entropy Search (MES) (Wang & Jegelka, 2017) to the DPMM-GP framework. MES seeks to evaluate the candidate point $x$ that maximizes the mutual information between the observation $y$ at $x$ and the global maximum value $y^* = \max_{x' \in \mathcal{X}} f(x')$.

The acquisition function is defined as the expected reduction in the entropy of the predictive distribution $p(y|x)$ induced by the knowledge of the global maximum $y^*$:

$$\alpha_{MES}(x) = I(y; y^*) = H(y|x) - \mathbb{E}_{y^*}\left[H(y \mid x, y < y^*)\right] \tag{24}$$

where the expectation is taken over the posterior distribution of the global maximum $p(y^*|\mathcal{D})$. Due to the multimodal nature of the DPMM-GP posterior, neither term has a closed analytical form. We derive tractable approximations for both below.

**Entropy of the Predictive Mixture**  The predictive distribution $p(y|x)$ is a Gaussian Mixture Model (GMM) with weights $w_k(x)$ and component parameters $\{\mu_{*,k}(x), \sigma_{*,k}^2(x)\}$ (Theorem 3.3). As the entropy of a GMM does not have a closed-form expression, we approximate it via *moment matching*. We treat the entropy of the mixture as the entropy of a single Gaussian with the equivalent variance $\sigma_{mix}^2(x)$ derived in Theorem 3.4:

$$H(y|x) \approx \frac{1}{2} \log\left(2\pi e \sigma_{mix}^2(x)\right). \tag{25}$$

This provides a coherent upper bound on the true entropy, as the Gaussian distribution maximizes entropy for a fixed variance.

**Expected Conditional Entropy**  The second term requires computing the entropy of the predictive distribution truncated at $y^*$, averaged over samples of $y^*$. We approximate the distribution $p(y^*)$ via Monte Carlo sampling. To draw a sample $y_s^*$, we use a two-stage ancestral sampling procedure consistent with our generative model:

1. **Regime Selection:** Sample a latent regime index $k \sim \text{Categorical}(\pi)$, where $\pi$ represents the global cluster weights.

2. **Function Maximization:** Draw a sample from the global maximum of the $k$-th GP component. Following standard practice, we approximate this via a Gumbel distribution sample based on the discrete maximum of the GP on the training data.

Given a sample $y_s^*$, the conditional distribution $p(y|x, y < y_s^*)$ is a truncated GMM. We approximate its entropy as the probabilistically weighted sum of the entropies of its truncated Gaussian components. Let $\gamma_{k,s} = \frac{y_s^* - \mu_{*,k}(x)}{\sigma_{*,k}(x)}$ be the standardized distance to the maximum for regime $k$. The entropy of the $k$-th Gaussian component truncated at $y_s^*$ is:

$$H_k(y|y < y_s^*) = \frac{1}{2} \log(2\pi e \sigma_{*,k}^2) + \ln \Phi(\gamma_{k,s}) - \frac{1}{2} \frac{\gamma_{k,s} \phi(\gamma_{k,s})}{\Phi(\gamma_{k,s})}. \tag{26}$$

Substituting this into Eq. (24) and simplifying (noting that the constant variance terms cancel out in the differential information gain formulation), we arrive at the numerical estimator:

$$\alpha_{MES}(x) \approx \frac{1}{S} \sum_{s=1}^{S} \sum_{k=1}^{K+1} w_k(x) \left[\frac{\gamma_{k,s} \phi(\gamma_{k,s})}{2\Phi(\gamma_{k,s})} - \ln \Phi(\gamma_{k,s})\right] \tag{27}$$

where $S$ is the number of Monte Carlo samples for $y^*$, and $\phi, \Phi$ are the standard normal PDF and CDF, respectively.

## B.2 Probability of Improvement (PI)

We extend the Probability of Improvement (PI) strategy (Kushner, 1964) to the DPMM-GP framework. Standard PI seeks to maximize the probability that the function value at a candidate point $x$ exceeds the current best observation $f^+$ by some margin $\xi \geq 0$.

In the context of our mixture model, the predictive distribution is multimodal. Consequently, the probability of improvement is not merely a function of a single mean and variance, but a weighted combination of the improvement probabilities offered by each latent regime.

**Theorem B.1** (DPMM-GP Probability of Improvement). *Let $f^+$ denote the current best observed value, and let $\xi \geq 0$ be a user-specified exploration parameter. The Probability of Improvement at input $x$ under the DPMM-GP posterior is the probability-weighted sum of the PI values for each constituent GP component:*

$$\alpha_{PI}(x) = \sum_{k=1}^{K+1} w_k(x) \cdot \Phi\left(\frac{\mu_{*,k}(x) - f^+ - \xi}{\sigma_{*,k}(x)}\right) \tag{28}$$

*where $w_k(x)$ are the spatially-modulated predictive weights defined in Eq. (9) (Proposition 3.4), and $(\mu_{*,k}(x), \sigma_{*,k}(x))$ are the posterior mean and standard deviation of the $k$-th GP regime.*

*Proof.* The acquisition function $\alpha_{PI}(x)$ is defined as the probability that the latent function value $f(x)$ exceeds the target $f^+ + \xi$:

$$\alpha_{PI}(x) = \mathbb{P}[f(x) > f^+ + \xi]. \tag{29}$$

We proceed by marginalizing over the latent regime assignments $z_*$ for the test point $x$. Using the Law of Total Probability:

$$\mathbb{P}[f(x) > f^+ + \xi] = \sum_{k=1}^{K+1} \mathbb{P}[f(x) > f^+ + \xi \mid z_* = k] \cdot p(z_* = k \mid x, \mathcal{D}). \tag{30}$$

From Theorem 3.3, the conditional distribution of $f(x)$ given the assignment $z_* = k$ is a Gaussian Process posterior:

$$p(f(x) \mid z_* = k) = \mathcal{N}(f(x) \mid \mu_{*,k}(x), \sigma_{*,k}^2(x)). \tag{31}$$

The conditional probability of improvement for this specific Gaussian component is given by the standard PI formula:

$$\mathbb{P}[f(x) > f^+ + \xi \mid z_* = k] = \int_{f^+ + \xi}^{\infty} \mathcal{N}(f \mid \mu_{*,k}(x), \sigma_{*,k}^2(x)) \, df \tag{32}$$

$$= \Phi\left(\frac{\mu_{*,k}(x) - (f^+ + \xi)}{\sigma_{*,k}(x)}\right). \tag{33}$$

Substituting the mixture weights $w_k(x)$ for $p(z_* = k \mid x, \mathcal{D})$ and the component probabilities back into the total probability sum yields Eq. (28). $\square$

## B.3 Upper Confidence Bound (UCB)

The Upper Confidence Bound (UCB) acquisition function (Srinivas et al., 2010) is widely used for its explicit management of the exploration-exploitation trade-off. Standard UCB is defined for a single Gaussian posterior; however, it extends naturally to the DPMM-GP by utilizing the mixture moments derived in Theorem 3.4.

Given the posterior predictive mixture with mean $\mu_{mix}(x)$ and variance $\sigma_{mix}^2(x)$, the Mixture UCB acquisition function is defined as:

$$\alpha_{UCB}(x) = \mu_{mix}(x) + \sqrt{\beta_t} \cdot \sigma_{mix}(x) \tag{34}$$

where $\beta_t$ is a time-dependent confidence parameter. A key property of RAMBO is that the variance term $\sigma_{mix}(x)$ encapsulates two distinct forms of uncertainty (Eq. 10):

- **Intra-regime uncertainty:** The average aleatoric variance of the individual GP experts ($\sum w_k \sigma_{*,k}^2$).

- **Inter-regime disagreement:** The epistemic variance of the means across different regimes ($\mathbb{V}ar_Z[\mu]$).

Consequently, Mixture UCB drives exploration not only where individual GPs are uncertain, but also where the regime assignment itself is ambiguous, naturally targeting regime boundaries for structural refinement.

## B.4 Thompson Sampling (TS)

Thompson Sampling (TS) (Thompson, 1933) is a randomized strategy that selects the next query point by optimizing a sample drawn from the posterior. It naturally handles the hierarchical structure of the DPMM-GP via ancestral sampling.

To select the next query point $x_{new}$, we perform the following two-stage procedure:

1. **Sample a Regime Assignment:** First, we sample a global regime index $k$ from the current categorical weights of the mixture components:

$$\hat{z} \sim \text{Categorical}(\pi_1, \ldots, \pi_K, \pi_{new}). \tag{35}$$

2. **Sample a Function Trajectory:** Conditioned on the chosen regime $\hat{z}$, we draw a continuous function realization $\tilde{f}(\cdot)$ from the corresponding Gaussian Process posterior $\mathcal{GP}(\mu_{*,\hat{z}}, \Sigma_{*,\hat{z}})$. In practice, this is approximated efficiently using Random Fourier Features (RFF).

3. **Optimization:** The next query point is the global maximizer of the sampled function:

$$x_{new} = \arg\max_{x \in \mathcal{X}} \tilde{f}(x). \tag{36}$$

As $t \to \infty$, the posterior probability of the true regime approaches 1, and TS asymptotically recovers the behavior of optimizing the correct underlying expert.

## B.5 Knowledge Gradient (KG)

The Knowledge Gradient (KG) acquisition function (Frazier et al., 2009) quantifies the expected one-step improvement in the global maximum of the posterior predictive mean. Unlike EI, which measures improvement over the best *observation*, KG values the improvement in the *model's estimate* of the optimum.

Let $\mu_n^* = \max_{x' \in \mathcal{X}} \mu_{mix,n}(x')$ denote the global maximum of the current mixture mean surface given dataset $\mathcal{D}_n$. If we were to effectively sample a candidate $(x, y)$, the dataset would evolve to $\mathcal{D}_{n+1} = \mathcal{D}_n \cup \{(x, y)\}$. This results in a new random posterior mean surface $\mu_{mix,n+1}(\cdot)$.

The KG acquisition value is defined as the expected increase in this surface maximum, marginalizing over the unknown outcome $y$ and the latent regime assignment of the new point:

$$\alpha_{KG}(x) = \mathbb{E}_{y|x,\mathcal{D}_n} \left[ \max_{x' \in \mathcal{X}} \mu_{mix,n+1}(x' \mid x, y) - \mu_n^* \right]. \tag{37}$$

**Posterior Mean Update (The Fantasization Process)** Computing $\mu_{mix,n+1}$ requires updating the DPMM-GP posterior with a "fantasy" observation $(x, y)$. In the exact inference limit, adding a point would require resampling all discrete regime assignments $z_{1:n}$. For computational tractability in the inner loop, we employ a *local update approximation*: we assume the assignments of the existing $n$ points remain fixed, and we update the model based on the probabilistic assignment of the new point $x$.

The updated mixture mean at any test point $x'$ is given by:

$$\mu_{mix,n+1}(x' \mid x, y) = \sum_{k=1}^{K+1} w_k^{(n+1)}(x') \cdot \mu_{*,k}^{(n+1)}(x'). \tag{38}$$

We compute the updated components as follows:

- **Component GP Update:** For each regime $k$, the GP posterior is updated conditioning on the event that $(x, y)$ belongs to regime $k$. The updated mean function $\mu_{*,k}^{(n+1)}(x')$ follows standard GP recursive equations:

$$\mu_{*,k}^{(n+1)}(x') = \mu_{*,k}^{(n)}(x') + \frac{k_k(x', x)}{k_k(x, x) + \sigma_{n,k}^2}(y - \mu_{*,k}^{(n)}(x)). \tag{39}$$

- **Gating Weight Update:** The mixture weights $w_k(x')$ (Eq. 8) depend on the cluster counts $n_k$ and the local predictive variance. The fantasy point $(x, y)$ updates the effective count $n_k$ by the probability that $x$ belongs to $k$: $\hat{\gamma}_k(x) \propto w_k^{(n)}(x) \cdot \mathcal{N}(y \mid \mu_{*,k}^{(n)}(x), \sigma_{*,k}^{2(n)}(x))$. The updated weights $w_k^{(n+1)}(x')$ are computed using the effective counts $n_k + \hat{\gamma}_k(x)$:

$$w_k^{(n+1)}(x') \propto \frac{n_k + \hat{\gamma}_k(x)}{n + 1 + \alpha} \cdot \exp\left(-\frac{1}{2}\log \sigma_{*,k}^{2(n+1)}(x')\right). \tag{40}$$

Since $\mu_{mix,n+1}(x')$ is non-convex, we evaluate Eq. (37) via Monte Carlo integration with $M$ fantasy samples drawn from the current posterior mixture $y^{(m)} \sim \sum w_k(x)\mathcal{N}(\mu_{*,k}(x), \sigma_{*,k}^2(x))$:

$$\alpha_{KG}(x) \approx \frac{1}{M} \sum_{m=1}^{M} \left(\max_{x' \in \mathcal{X}} \mu_{mix,n+1}(x' \mid x, y^{(m)}) - \mu_n^*\right). \tag{41}$$

The inner maximization is solved via multi-start L-BFGS, utilizing the updated gradients of the mixture mean.

## B.6 Predictive Entropy Search (PES)

Predictive Entropy Search (PES) (Hernández-Lobato et al., 2014) maximizes the mutual information between the observation $y$ and the *location* of the global optimizer $x^*$.

$$\alpha_{PES}(x) = H(y|x, \mathcal{D}_n) - \mathbb{E}_{x^*|\mathcal{D}_n}\left[H(y \mid x, \mathcal{D}_n, x^*)\right]. \tag{42}$$

The first term is the entropy of the current posterior predictive mixture. As exact entropy for GMMs is intractable, we use the moment-matched Gaussian approximation (Theorem 3.4):

$$H(y|x, \mathcal{D}_n) \approx \frac{1}{2}\log\left(2\pi e \sigma_{mix}^2(x)\right). \tag{43}$$

The second term is the expected entropy of $y$ conditioned on the constraint that $x^*$ is the global maximizer. This constraint implies $f(x^*) \geq f(x)$ for all $x$. We approximate this intractable expectation using a two-step procedure:

We draw samples of the global optimizer location utilizing the hierarchical generative process of the DPMM-GP. To generate a sample $x_s^*$:

- Sample a regime assignment $\hat{z} \sim \text{Categorical}(\{w_k(x)\}_{k=1}^K)$.

- Conditioned on $\hat{z}$, sample a function path $\tilde{f} \sim \mathcal{GP}(\mu_{*,\hat{z}}, \Sigma_{*,\hat{z}})$ using Random Fourier Features (RFF) to ensure differentiability.

- Optimize the sampled path: $x_s^* = \arg\max_{x \in \mathcal{X}} \tilde{f}(x)$.

Conditioning on $x_s^*$ imposes a complex set of gradient and value constraints on the random variable $y(x)$. For computational feasibility, we approximate the conditioning $p(y \mid x, x_s^*)$ by the necessary condition $y(x) < y(x_s^*) \approx \tilde{f}(x_s^*)$. This effectively truncates the predictive distribution at the sampled maximum value $f_{max}^s = \tilde{f}(x_s^*)$.

The conditional entropy is approximated as the entropy of the mixture distribution truncated at $f_{max}^s$. For a single component $k$, the truncated variance $v_k$ given an upper truncation point $\beta$ is:

$$v_k(\beta) = \sigma_{*,k}^2(x)\left[1 - \delta_k(\beta)(\delta_k(\beta) + \alpha_k(\beta))\right] \tag{44}$$

where $\alpha_k(\beta) = \frac{\beta - \mu_{*,k}(x)}{\sigma_{*,k}(x)}$ and $\delta_k(\beta) = \frac{\phi(\alpha_k(\beta))}{\Phi(\alpha_k(\beta))}$.

Aggregating over the mixture weights $w_k(x)$, the expected conditional entropy is approximated by:

$$\mathbb{E}_{x^*}[H(y|x, x^*)] \approx \frac{1}{S} \sum_{s=1}^{S} \frac{1}{2} \log \left( 2\pi e \sum_{k=1}^{K} w_k(x) v_k(f_{max}^s) \right). \tag{45}$$

This formulation directs sampling to regions where the outcome $y$ would most significantly constrain the possible locations of $x^*$.

# C   Algorithms

## C.1   Collapsed Gibbs Sampler for DPMM-GP

Algorithm 1 presents the collapsed Gibbs sampler used for posterior inference in DPMM-GP. The key insight enabling efficient sampling is the analytical marginalization of latent function values $\mathbf{f}$, reducing the state space to only cluster assignments $\mathbf{z}$ and hyperparameters $\Theta$.

The sampler iterates over each observation, temporarily removing it from its current cluster (lines 6–7) and computing reassignment probabilities. For existing clusters, the probability is proportional to the cluster size weighted by the GP predictive likelihood (line 8); for a new cluster, it is proportional to $\alpha$ weighted by the prior predictive density under $G_0$ (line 9). This follows directly from the Chinese Restaurant Process. After reassignment (lines 10–12), hyperparameters for each active cluster are updated either via Metropolis-Hastings or gradient ascent on the marginal likelihood (lines 14–15).

---

**Algorithm 1** DPMM-GP Collapsed Gibbs Sampler

---

1: **Input:** Data $\mathcal{D}$, iterations $T$, conc. $\alpha$, base $G_0$
2: Initialize $z_i$ randomly for $i = 1 \ldots n$
3: Initialize $\theta_k \sim G_0$ for initial clusters
4: **for** $t = 1$ **to** $T$ **do**
5:   **for** $i = 1$ **to** $n$ **do**
6:     Remove $i$ from current cluster: $n_{z_i} \leftarrow n_{z_i} - 1$
7:     If cluster empty, remove $\theta_{z_i}$
8:     Compute probs for existing clusters: $p_k \propto n_{k,-i} \cdot \mathcal{N}(y_i \mid \mathcal{D}_{k,-i}, \theta_k)$         ▷ Excluding observation $i$
9:     Compute prob for new cluster: $p_{\text{new}} \propto \alpha \cdot p(y_i \mid \mathbf{x}_i, G_0)$
10:     Sample $z_i \sim \text{Categorical}(p_1, \ldots, p_K, p_{\text{new}})$
11:     If $z_i = \text{new}$, draw $\theta_{\text{new}} \sim p(\theta \mid y_i, G_0)$
12:     Add $i$ to new cluster: $n_{z_i} \leftarrow n_{z_i} + 1$
13:   **end for**
14:   **for** each active cluster $k$ **do**
15:     Update $\theta_k$ via MH or Gradient Ascent
16:   **end for**
17: **end for**

---

## C.2 Optimization Loop of RAMBO

RAMBO integrates the DPMM-GP surrogate with a sequential acquisition strategy, as outlined in Algorithm 2. The procedure begins by initializing 20 identical quasirandom Sobol points to ensure space-filling coverage. At each iteration, we perform collapsed Gibbs inference with warm starts from the previous assignments and hyperparameters to reduce burn-in overhead. The mixture Expected Improvement is then maximized via multi-start L-BFGS-B, initialized from uniform random samples, regime centroids, and perturbations around the current best solution. Finally, we update the dataset with the new observation and prune empty or low-weight regimes to maintain computational efficiency.

---

**Algorithm 2** RAMBO: Regime-Adaptive Mixture Bayesian Optimization

---

**Require:** Search space $\mathcal{X}$, budget $T_{\max}$, initial size $n_{\text{init}}$
**Require:** Base concentration $\alpha_0$, MCMC samples $S$, restarts $R$, prune threshold $\epsilon$
**Ensure:** Best solution $\mathbf{x}^*$
 1: **// Phase 1: Initialization**
 2: $\mathcal{D}_0 \leftarrow \text{SOBOL}(\mathcal{X}, n_{\text{init}})$
 3: Initialize $\Theta^{(0)}$ and $\mathbf{z}^{(0)}$ randomly
 4: $\mathbf{x}^* \leftarrow \arg\max_{(\mathbf{x},y)\in\mathcal{D}_0} y$
 5: **for** $t = 1$ **to** $T_{\max}$ **do**
 6:    **// Phase 2: Adaptive $\alpha$-Scheduling**
 7:    $\alpha_t \leftarrow \alpha_0 \cdot \frac{\sqrt{t}}{\log(t+e)}$                                          $\triangleright$ Log-Sqrt Schedule (Eq. 15)
 8:    **// Phase 3: Inference (Warm Start)**
 9:    Run Collapsed Gibbs (Alg. 1) for $S$ steps with $\alpha_t$, initialized from $(\Theta^{(t-1)}, \mathbf{z}^{(t-1)})$
10:    Collect post-burn-in posterior samples $\{(\Theta^{(s)}, \mathbf{z}^{(s)})\}_{s=1}^{S}$          $\triangleright$ Used for acquisition only
11:    Compute mixture moments $\mu_{\text{mix}}(\mathbf{x}), \sigma_{\text{mix}}(\mathbf{x})$ and weights $w_k(\mathbf{x})$ as the sample-mean of per-sample (Thm. 3.5) over $s = 1, \ldots, S$
12:    **// Phase 4: Acquisition Optimization**
13:    Define $\alpha_{\text{EI}}(\mathbf{x})$ per Eq. (13)
14:    Generate start points $\mathcal{S}_{\text{init}} \leftarrow \{\text{UNIFORM}(\mathcal{X})\} \cup \{\text{CENTROIDS}(\mathbf{z})\} \cup \{\mathbf{x}^* + \boldsymbol{\delta}\}$
15:    $\mathbf{x}_{\text{new}} \leftarrow \arg\max_{\mathbf{x}\in\mathcal{S}_{\text{init}}} \text{L-BFGS-B}(\alpha_{\text{EI}}(\mathbf{x}))$
16:    **// Phase 5: Evaluation & Update**
17:    $y_{\text{new}} \leftarrow f(\mathbf{x}_{\text{new}}) + \varepsilon$
18:    $\mathcal{D}_t \leftarrow \mathcal{D}_{t-1} \cup \{(\mathbf{x}_{\text{new}}, y_{\text{new}})\}$
19:    **if** $y_{\text{new}} > f(\mathbf{x}^*)$ **then**
20:       $\mathbf{x}^* \leftarrow \mathbf{x}_{\text{new}}$
21:    **end if**
22:    **// Phase 6: Maintenance**
23:    Update $(\Theta^{(t)}, \mathbf{z}^{(t)})$ using the last MCMC sample
24:    Prune regimes where $\sum_i \mathbb{I}(z_i = k) < 1$ or $\pi_k < \epsilon$
25: **end for**

---

## D    Benchmarks

### D.1    Levy Function

The Levy function is characterized by a rugged surface with a high density of local minima, designed to challenge an optimizer's ability to resolve fine-grained structures. In the $d$-dimensional case, the function is defined as:

$$f(\mathbf{x}) = \sin^2(\pi w_1) + \sum_{i=1}^{d-1}(w_i - 1)^2[1 + 10\sin^2(\pi w_i + 1)] + (w_d - 1)^2[1 + \sin^2(2\pi w_d)]$$

where $w_i = 1 + \frac{x_i - 1}{4}$ for $i = 1, \ldots, d$.

For our 2D visualization and validation, we consider $x_i \in [-10, 10]$. The landscape exhibits intense high-frequency oscillations (as shown in Figure 2), with the global minimum $f(\mathbf{x}^*) = 0$ located at the interior point $\mathbf{x}^* = (1, \ldots, 1)$.

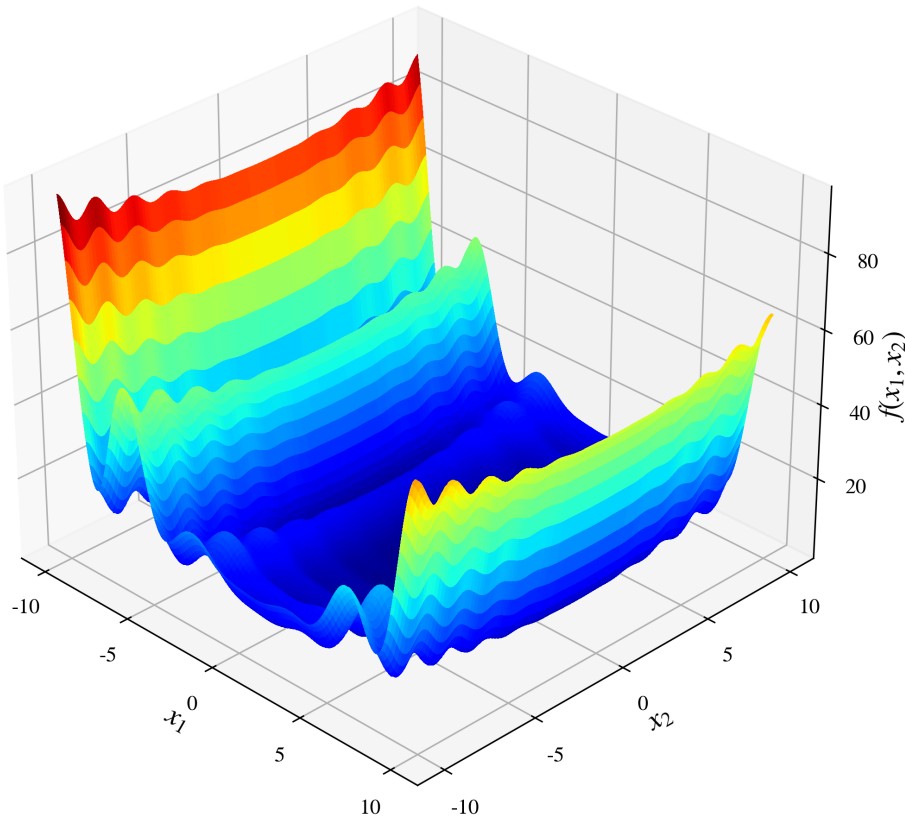

*Figure 2.* 3D landscape of the 2D Levy function, illustrating the dense clusters of local minima and rugged surface topology.

### D.2    Schwefel Function

The Schwefel function presents a deceptive landscape where the global optimum is geometrically isolated near the domain boundaries. This structure is particularly difficult for stationary Gaussian Processes, as it penalizes methods that bias their search toward the central region of the domain. The mathematical representation is given by:

$$f(\mathbf{x}) = 418.9829d - \sum_{i=1}^{d} x_i \sin(\sqrt{|x_i|})$$

defined over the hypercube $x_i \in [-500, 500]$. As visualized in Figure 3, the function contains numerous sub-optimal peaks and basins.

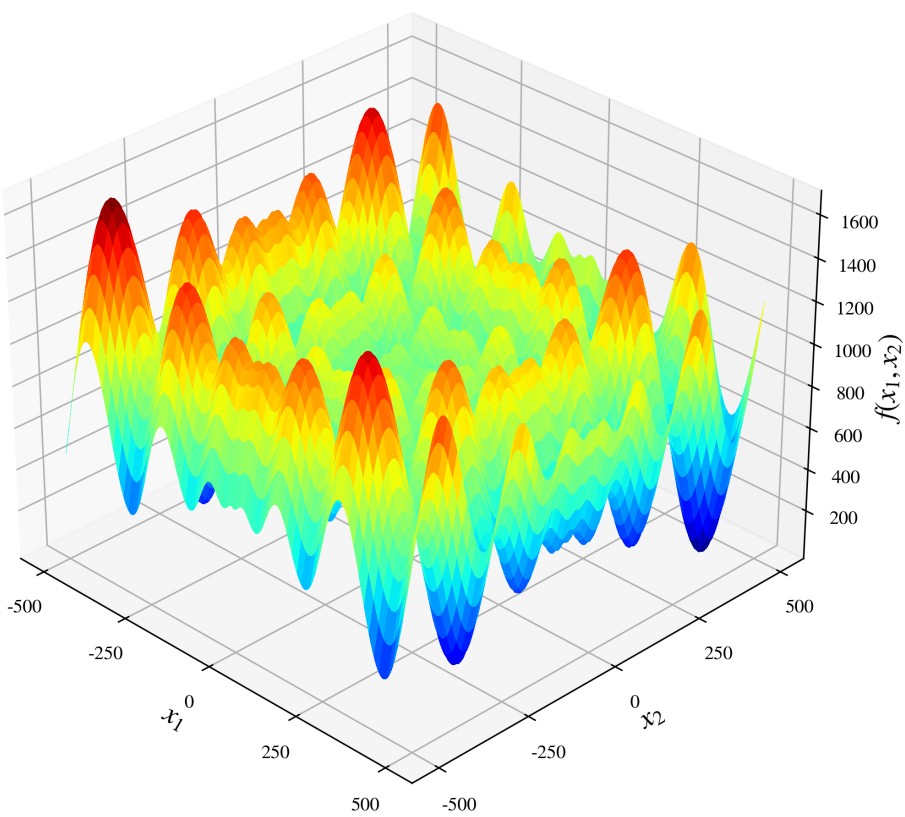

*Figure 3.* 3D landscape of the 2D Schwefel function, showcasing the deceptive local optima and the isolated nature of the global minimum.

### D.3 Molecular Conformer Optimization (12D)

The Molecular Conformer Optimization task serves as a high-dimensional, real-world benchmark to evaluate the scalability of RAMBO in pathological multi-modal landscapes. This problem involves finding the global minimum energy configuration of a pentadecane chain ($C_{15}H_{32}$), where the state space is defined by $d = 12$ internal dihedral (torsion) angles $\boldsymbol{\theta} = (\theta_1, \ldots, \theta_{12})$.

**Mathematical Representation**    The objective is to minimize the total potential energy $E(\boldsymbol{\theta})$ of the conformer. We model this energy using the **MMFF94 force field** (Halgren, 1996), which can be decomposed into torsional contributions and non-bonded interactions:

$$E(\boldsymbol{\theta}) = \sum_{i=1}^{12} V_{tors}(\theta_i) + E_{non-bonded}(\mathbf{R}) \tag{46}$$

where $V_{tors}(\theta_i)$ typically follows a periodic potential:

$$V_{tors}(\theta) = \sum_{n=1}^{3} \frac{V_n}{2}[1 + \cos(n\theta - \gamma_n)] \tag{47}$$

The rotation around each $C - C$ bond favors the $180°$ (anti) and $\pm 60°$ (gauche) orientations. This discrete preference

induces a combinatorial explosion of $3^{12} = 531,441$ potential conformational minima. These stable states are separated by high-energy steric barriers, creating a landscape characterized by sharp transitions and high-frequency oscillations.

**Implementation Details**  To evaluate a configuration $\theta$, we construct the molecular backbone using **RDKit** (Landrum et al., 2013). Crucially, to ensure a realistic energy landscape, we perform a **constrained geometry optimization**: the target dihedral angles $\theta$ are fixed using constraints, while all other degrees of freedom (bond lengths, angles, and Cartesian coordinates $\mathbf{R}$) are relaxed to minimize the energy. This formulation prevents unrealistic steric clashes characteristic of rigid-body rotation and ensures the optimizer navigates a chemically valid potential energy surface.

**Problem Visualization**  Figure 4 illustrates the geometric definition of a dihedral angle within the molecular chain, defined by a sequence of four consecutively bonded atoms (labeled Atoms 1–4). The rotation occurs around the central bond connecting Atom 2 and Atom 3, which serves as the rotation axis. This rotational motion determines the relative orientation of two intersecting planes: Plane 1, defined by Atoms 1, 2, and 3, and Plane 2, defined by Atoms 2, 3, and 4. The dihedral angle is the angle between these two planes.

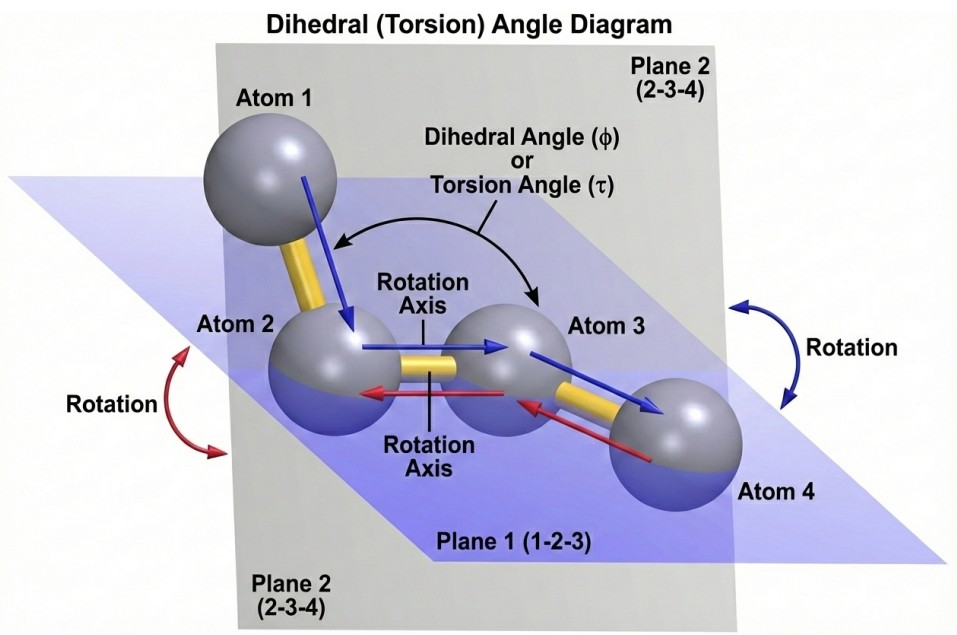

*Figure 4.* Dihedral (Torsion) Angle Diagram: The optimization space consists of 12 such rotational degrees of freedom, where steric hindrance between atoms 1 and 4 creates complex energy barriers.

### D.4  Virtual Screening for Drug Discovery (Cancer-6T2W, 50D)

We evaluate RAMBO on a drug discovery benchmark targeting the Colony-Stimulating Factor 1 Receptor (CSF1R) kinase domain (PDB ID: 6T2W) (Goldberg et al., 2020). CSF1R is a type III receptor tyrosine kinase that regulates tumor-associated macrophage differentiation and survival; its overexpression correlates with poor prognosis across multiple cancer types including breast, ovarian, and lung carcinomas (Wen et al., 2023). Pharmacological inhibition of CSF1R has emerged as a promising immunotherapeutic strategy, with pexidartinib (PLX3397) receiving FDA approval in 2019 for tenosynovial giant cell tumor (Tap et al., 2015).

The benchmark dataset, derived from the DrugImprover framework (Liu et al., 2023), comprises 1 million small molecules sampled from the ZINC15 database (Sterling & Irwin, 2015), each annotated with docking scores computed using the OpenEye FRED software (McGann, 2012). The BO objective is to minimize the docking score (more negative values indicate stronger predicted binding affinity). To enable continuous optimization, we represent molecules via 2048-bit Morgan fingerprints (Rogers & Hahn, 2010) compressed to $d = 50$ dimensions using Principal Component Analysis, following standard practice in latent-space molecular optimization (Gómez-Bombarelli et al., 2018).

This benchmark poses two challenges characteristic of real-world drug discovery: (1) **high dimensionality**—the 50D

latent space necessitates efficient exploration strategies; and (2) **multi-regime structure**—distinct molecular scaffolds (e.g., different ring systems, functional groups, or pharmacophores) occupy disjoint regions of the latent space with fundamentally different structure-activity relationships (SAR). A single stationary GP cannot capture these scaffold-dependent landscapes, as binding affinity varies non-smoothly across chemical families. RAMBO's regime-adaptive mechanism naturally partitions the chemical space into scaffold-specific clusters, enabling locally accurate surrogate modeling within each chemical series.

## D.5 Nuclear Fusion Reactor Design (ConStellaration, 80D)

We evaluate RAMBO on the ConStellaration benchmark (Cadena et al., 2025), a recently released dataset and optimization challenge for quasi-isodynamic (QI) stellarator design developed by Proxima Fusion in collaboration with Hugging Face. Stellarators are magnetic confinement devices that represent a promising path toward steady-state, disruption-free fusion energy (Gates et al., 2018; Helander, 2014). Unlike tokamaks, stellarators rely entirely on external electromagnetic coils to confine the plasma, avoiding current-driven instabilities but requiring complex three-dimensional magnetic field geometries that must be carefully optimized (Goodman et al., 2023).

**Dataset and Representation.** The ConStellaration dataset comprises approximately 158,000 QI-like stellarator plasma boundary configurations, each paired with ideal magnetohydrodynamic (MHD) equilibria computed using VMEC++ (Hirshman & Whitson, 1983) and associated performance metrics. The plasma boundary is parameterized by a truncated Fourier series in cylindrical coordinates $(R, Z)$:

$$R(\theta, \phi) = \sum_{m,n} R_{mn} \cos(m\theta - nN_{\text{fp}}\phi), \quad Z(\theta, \phi) = \sum_{m,n} Z_{mn} \sin(m\theta - nN_{\text{fp}}\phi),$$

where $\theta$ and $\phi$ are poloidal and toroidal angles, and $N_{\text{fp}}$ is the number of field periods. Assuming stellarator symmetry (i.e., $R(\theta, \phi) = R(-\theta, -\phi)$ and $Z(\theta, \phi) = -Z(-\theta, -\phi)$) and fixing the major radius $R_{0,0} = 1$, the optimization problem has $d = 80$ degrees of freedom corresponding to the Fourier coefficients $\{R_{mn}, Z_{mn}\}$.

**Optimization Objective.** We adopt the "simple-to-build QI stellarator" benchmark, which seeks to maximize the quasi-isodynamic quality metric $Q_{\text{QI}}$ while satisfying constraints on aspect ratio, rotational transform, and coil complexity. The QI property ensures that trapped particle orbits have vanishing average radial drift, which is critical for minimizing neoclassical transport and eliminating bootstrap currents that can destabilize the plasma (Helander & Nührenberg, 2009). Formally, the objective combines multiple physics targets including: (1) minimization of the effective ripple $\epsilon_{\text{eff}}$ that governs neoclassical transport; (2) enforcement of poloidally closed $|\mathbf{B}|$ contours characteristic of QI fields; and (3) penalization of high-curvature boundary shapes that would require complex coils to reproduce.

**Multi-Regime Structure.** Stellarator design exhibits highly nonlinear physics with abrupt transitions between qualitatively different magnetic field topologies. Small perturbations in boundary shape can trigger transitions between nested flux surfaces and magnetic islands, between regions of good and poor particle confinement, or between MHD-stable and unstable configurations. The optimization landscape is thus characterized by disconnected basins corresponding to fundamentally different plasma geometries—configurations with different numbers of field periods (1, 2, 3, 4, or 5 in the dataset), different magnetic well depths, and different elongation profiles occupy distinct regions of the Fourier coefficient space with incommensurable local curvature. This patchy landscape, where smooth regions of high confinement quality are interspersed with sharp transitions to unstable configurations, exemplifies the multi-regime structure that RAMBO is designed to capture. A single stationary GP with global hyperparameters cannot simultaneously model the smooth variation within each topology class and the abrupt transitions between them.

**Computational Cost.** Each function evaluation requires solving the 3D ideal-MHD equilibrium equations via VMEC++, which takes $\mathcal{O}(\text{seconds to minutes})$ depending on resolution, making this a genuinely expensive black-box optimization problem well-suited for Bayesian optimization.

# E    Additional Experimental Results

This appendix collects the additional empirical analyses promised in the rebuttal: computational overhead, ablation of adaptive $\alpha$-scheduling, comparison against a recently proposed vanilla-BO baseline, regression-quality evaluation of the DPMM-GP surrogate independent of the BO loop, and per-regime hyperparameter diagnostics that illustrate the regime structure RAMBO discovers.

## E.1    Computational Overhead (Wall-Clock Time)

Table 1 reports per-iteration wall-clock time on the Levy-6D benchmark, measured on a single CPU. RAMBO's collapsed Gibbs sampler is the dominant cost—the per-iteration surrogate-fitting time is 32.27 seconds, considerably higher than SAASBO's 18.10 seconds and an order of magnitude higher than TuRBO. We emphasize, however, that these numbers measure only surrogate fitting and do not include the function evaluation cost. For the scientific design problems targeted by RAMBO (molecular conformer optimization with MMFF94 force field, drug discovery with docking simulation, stellarator equilibria via VMEC++), a single function evaluation typically takes seconds to minutes, so the surrogate overhead is negligible relative to the cost of the objective. The improved sample efficiency of RAMBO—reaching high-quality solutions with substantially fewer evaluations—therefore translates directly into lower total wall-clock time on the regime targeted by the method. Reducing the MCMC overhead via variational approximations remains a promising direction for future work.

*Table 1.* Per-iteration wall-clock time (Levy-6D, single CPU, surrogate fitting only).

| Method | Time/iter (sec) |
|---|---|
| SMAC | 0.140 |
| Vanilla BO | 0.290 |
| Bounce | 0.375 |
| TuRBO | 0.542 |
| ALEBO | 0.545 |
| BAxUS | 0.738 |
| SGP | 1.979 |
| HEBO | 2.665 |
| COMBO | 7.383 |
| SAASBO | 18.095 |
| **RAMBO** | **32.269** |

## E.2    Ablation: Adaptive $\alpha$-Scheduling

Table 2 compares the Log-Sqrt adaptive schedule against fixed $\alpha$ values on Levy-6D. The optimal fixed $\alpha$ varies across benchmarks (e.g., $\alpha = 0.5$ is the best fixed choice on Levy-6D, but $\alpha = 1.0$ would be preferred on some other landscapes), and poor choices induce instability—$\alpha = 2.0$ has a standard error of $0.578$, more than six times that of the adaptive schedule. The Log-Sqrt schedule provides a principled, benchmark-independent default that is competitive with the best fixed choice and avoids the catastrophic-failure mode of over-large $\alpha$.

*Table 2.* Final-iteration best objective (Levy-6D, lower is better) under fixed vs. adaptive $\alpha$. Mean $\pm$ standard error over 5 seeds.

| Setting | Final best (Levy-6D) |
|---|---|
| Fixed $\alpha = 0.5$ | $-0.319 \pm 0.063$ |
| Fixed $\alpha = 1.0$ | $-0.276 \pm 0.053$ |
| Fixed $\alpha = 2.0$ | $-0.653 \pm 0.578$ |
| Fixed $\alpha = 5.0$ | $-0.403 \pm 0.154$ |
| **Adaptive (Log-Sqrt)** | $\mathbf{-0.238 \pm 0.094}$ |

## E.3    Comparison Against Vanilla BO with Dimension-Dependent Length-Scale Priors

Recent work (Hvarfner et al., 2024) suggests that vanilla BO with dimension-dependent length-scale priors can rival specialized high-dimensional methods. Table 3 reports a head-to-head comparison on all seven benchmarks. Vanilla BO closes some of the gap to specialized methods on smooth, low-dimensional landscapes but degrades significantly on multi-regime objectives (Levy-10D, Schwefel-6D/10D, Molecular Conformer, ConStellaration), where RAMBO outperforms by

an order of magnitude. This confirms that, while priors are an effective remedy for high-dimensionality alone, they do not address the multi-regime structure that RAMBO is designed for.

*Table 3.* Final-iteration best objective (mean $\pm$ standard error over 5 seeds). Lower is better for Levy, Schwefel, Molecular Conformer; higher is better for Drug Discovery and ConStellaration.

| Benchmark | Vanilla BO (Hvarfner et al., 2024) | **RAMBO** | SGP |
|---|---|---|---|
| Levy-6D | $-0.870 \pm 0.089$ | $\mathbf{-0.238 \pm 0.094}$ | $-1.154 \pm 0.519$ |
| Levy-10D | $-17.009 \pm 11.698$ | $\mathbf{-2.095 \pm 0.606}$ | $-4.060 \pm 0.446$ |
| Schwefel-6D | $-1280.188 \pm 167.773$ | $\mathbf{-495.102 \pm 107.224}$ | $-1192.102 \pm 69.720$ |
| Schwefel-10D | $-2570.130 \pm 0.000$ | $\mathbf{-1420.399 \pm 331.557}$ | $-2555.652 \pm 22.865$ |
| Molecular Conformer | $-48.285 \pm 3.372$ | $\mathbf{-8.051 \pm 1.675}$ | $-66.740 \pm 20.267$ |
| Drug Discovery | $12.009 \pm 0.482$ | $\mathbf{13.309 \pm 0.514}$ | $12.501 \pm 0.000$ |
| ConStellaration | $0.120 \pm 0.081$ | $\mathbf{0.227 \pm 0.026}$ | $0.013 \pm 0.000$ |

## E.4 Surrogate Regression Quality (Independent of the BO Loop)

To disentangle surrogate modeling from acquisition design, we evaluate the DPMM-GP surrogate as a stand-alone regressor on three synthetic targets with explicit multi-regime structure and two UCI tabular regression benchmarks. We report 5-fold cross-validated RMSE.

**Synthetic targets.**

- **Piecewise-Smooth** ($d = 5$): two regimes split by $x_0 = 0$. Low-frequency $y = 2\sin(0.5x_0)$ for $x_0 < 0$, and a rough high-frequency regime with $\varepsilon \sim \mathcal{N}(0, 0.05^2)$ for $x_0 \geq 0$.

- **Heteroscedastic** ($d = 4$): identical functional form $y = \sin(x_0) + 0.5x_1 + 0.3x_3$, but drastically different noise: $\sigma = 0.02$ for $x_0 > 0$ vs. $\sigma = 0.5$ for $x_0 \leq 0$.

- **Multi-Scale** ($d = 6$): high-frequency oscillations in the interior $|x_0| < 1.5$ and a smooth quadratic exterior, mimicking the multi-regime scientific landscapes RAMBO targets.

*Table 4.* 5-fold CV RMSE (300 training samples on synthetic targets; UCI dataset sizes shown). Lower is better.

| Method | Piecewise-Smooth | Heteroscedastic | Multi-Scale | Diabetes (442) | Energy (768) |
|---|---|---|---|---|---|
| SGP | $0.4036 \pm 0.0724$ | $0.4467 \pm 0.0723$ | $0.6922 \pm 0.1095$ | $56.750 \pm 3.013$ | $1.436 \pm 0.398$ |
| MoE-GP | $0.3587 \pm 0.0230$ | $0.4282 \pm 0.0758$ | $0.5679 \pm 0.0383$ | $56.522 \pm 2.311$ | $1.819 \pm 1.407$ |
| **DPMM-GP** | $\mathbf{0.3564 \pm 0.0260}$ | $\mathbf{0.3744 \pm 0.0482}$ | $\mathbf{0.5629 \pm 0.0394}$ | $\mathbf{56.171 \pm 3.094}$ | $\mathbf{0.957 \pm 0.045}$ |

Across all five targets, the DPMM-GP surrogate matches or exceeds both a standard SGP and a finite Mixture-of-Experts GP, with the largest gains on heteroscedastic and multi-scale landscapes—precisely the regime where RAMBO's regime-adaptive mechanism is intended to operate. The result confirms that the BO improvements reported in the main text are not an artifact of acquisition design alone.

## E.5 Per-Regime Hyperparameters Discovered by RAMBO

To illustrate the regime structure RAMBO discovers, Tables 5 and 6 report the per-regime kernel hyperparameters at the final iteration on Schwefel-10D and Molecular Conformer Optimization, alongside the global hyperparameters fit by a standard SGP on the same data.

*Table 5.* Schwefel-10D: per-regime hyperparameters discovered by RAMBO vs. global SGP hyperparameters (final iteration, $n = 219$).

| Regime | $\ell_k$ | $\sigma_{f,k}^2$ | $\sigma_{n,k}^2$ | #Points |
|---|---|---|---|---|
| 1 | 0.326 | 0.365 | 0.003 | 159 |
| 2 | 2.388 | 1.560 | 0.004 | 59 |
| 3 | 0.376 | 0.166 | 0.006 | 1 |
| SGP (global) | 241.144 | 352518.696 | 269153.624 | 219 |

*Table 6.* Molecular Conformer: per-regime hyperparameters discovered by RAMBO vs. global SGP hyperparameters (final iteration, $n = 219$).

| Regime | $\ell_k$ | $\sigma^2_{f,k}$ | $\sigma^2_{n,k}$ | #Points |
|--------|----------|------------------|------------------|---------|
| 1 | 98.865 | 0.092 | 0.000 | 27 |
| 2 | 279.259 | 1.138 | 0.000 | 22 |
| 3 | 39.172 | 0.077 | 0.000 | 37 |
| 4 | 20.433 | 0.073 | 0.000 | 49 |
| 5 | 2.200 | 0.098 | 0.003 | 66 |
| 6 | 2.170 | 0.557 | 0.008 | 15 |
| 7 | 0.324 | 0.120 | 0.005 | 3 |
| SGP (global) | 108.079 | 67275.802 | 9546.095 | 219 |

Two observations are noteworthy. (i) The regime hyperparameters exhibit clear divergence: on Schwefel-10D, RAMBO discovers a short-scale regime ($\ell = 0.326$) alongside a broad smooth regime ($\ell = 2.388$), with signal variance differing by a factor of $4\times$. The Molecular Conformer benchmark shows even sharper separation—length scales span two orders of magnitude across the 7 discovered regimes, consistent with conformational basins separated by sharp torsional barriers. (ii) The standard SGP's global hyperparameters are pathologically large ($\sigma^2_f > 10^5$, $\sigma^2_n > 10^4$ on Schwefel-10D)—the model is unable to fit heterogeneous hyperparameters and is forced into a degenerate compromise. This diagnostic directly visualizes the failure mode that RAMBO is designed to remedy.

# F  Adaptive Concentration Parameter Scheduling

The concentration parameter $\alpha$ governs regime creation in the Dirichlet Process: larger $\alpha$ encourages more clusters, with $\mathbb{E}[K_n \mid \alpha] = \alpha \log(n/\alpha + 1) + O(1)$ as $n \to \infty$ (Antoniak, 1974). Prior work either fixes $\alpha$ or learns it via MCMC (Rasmussen & Ghahramani, 2001), but for sequential optimization, the appropriate $\alpha$ varies with data availability. With few observations, the data cannot reliably distinguish true regime structure from noise, and premature fragmentation leaves each GP expert with insufficient data for stable hyperparameter estimation. As observations accumulate, finer regime structure becomes statistically identifiable. This motivates a principled scheduling strategy: start conservative, then allow complexity to grow.

We formalize this intuition by matching the prior's expected complexity to a target regime discovery rate.

**Proposition F.1.** *Assume the number of discernible regimes $K^*$ grows with sample size $n$ at a polynomial rate $\mathcal{O}(n^\beta)$ for $\beta \in (0,1)$. To align the Dirichlet Process prior expectation $\mathbb{E}[K_n \mid \alpha]$ with this target rate, the concentration parameter must scale as:*

$$\alpha^*(n) \propto \frac{n^\beta}{\log n}. \tag{48}$$

*Proof.* Under the Chinese Restaurant Process representation, the expected number of clusters after $n$ observations satisfies $\mathbb{E}[K_n \mid \alpha] = \alpha \log(1 + n/\alpha) + O(1)$ as $n \to \infty$ (Antoniak, 1974). Suppose we seek a schedule $\alpha_n$ such that $\mathbb{E}[K_n \mid \alpha_n] \asymp n^\beta$. Hypothesizing $\alpha_n = c \cdot n^\beta / \log n$ for some constant $c > 0$ and substituting,

$$\mathbb{E}[K_n \mid \alpha_n] = \frac{cn^\beta}{\log n} \log\left(1 + \frac{n^{1-\beta} \log n}{c}\right) + O(1). \tag{49}$$

As $n \to \infty$, the argument of the outer logarithm is dominated by $n^{1-\beta}$, so $\log(1 + n^{1-\beta} \log n/c) = (1 - \beta) \log n + O(\log \log n)$. Therefore

$$\mathbb{E}[K_n \mid \alpha_n] \sim c(1 - \beta)\, n^\beta, \tag{50}$$

which matches the target growth rate $\Theta(n^\beta)$. $\qquad\square$

Guided by Proposition F.1, we adopt a square-root growth assumption ($\beta = 1/2$), motivated by the observation that distinct basins of attraction in multi-modal landscapes are discovered at a rate analogous to Heap's Law in information retrieval or the "square-root rule" in clustering heuristics (Heaps, 1978). This yields the **Log-Sqrt Schedule**:

$$\alpha_t = \alpha_0 \cdot \frac{\sqrt{t}}{\log(t + e)}, \tag{51}$$

where $\alpha_0$ is a base concentration parameter (default $\alpha_0 = 1.0$) and the offset $e$ in the denominator ensures numerical stability at $t = 1$.

This schedule provides three desirable properties: (i) early parsimony—small $\alpha_t$ initially prevents premature fragmentation when observations are sparse; (ii) progressive refinement—increasing $\alpha_t$ enables fine-grained regime discovery as evidence accumulates; and (iii) bias-variance balance—the sub-linear growth rate avoids over-segmentation while permitting sufficient model flexibility. When $\alpha_t$ changes between iterations, we warm-start Gibbs sampling from the previous assignments $\mathbf{z}_{t-1}$ to preserve learned structure.

