# OpenReview forum: "Regime-Adaptive Bayesian Optimization via Dirichlet Process Mixtures of Gaussian Processes"
_ICML.cc/2026/Conference — ICML 2026 regular_

### Official Review · Reviewer_sirv · 2026-02-25

**Soundness:** 3
**Presentation:** 2
**Significance:** 3
**Originality:** 3
**Overall Recommendation:** 4
**Confidence:** 3

**Summary:**

A standard Gaussian process (GP) prior with a stationary kernel assumes that the function has the same characteristics across the entire domain -- which seldom is true for real-world functions. The paper proposes the Dirichlet Process Mixture of Gaussian Processes (DPMM-GP) to model functions with varying local characteristics. The DPMM-GP is combined with Expected Improvement (EI) and collapsed Gibbs sampling for Bayesian optimization. The proposed method is evaluated on synthetic and scientific application benchmarks and demonstrates strong performance.

**Compliance With Llm Reviewing Policy:**

Affirmed.

**Final Justification:**

The rebuttal addressed most of my concerns. However, the derivation of Eq. (8) that the authors provided seems to be incorrect (although this is not critical since it is mostly a design choice), see updated rebuttal acknowledgement.

**Key Questions For Authors:**

- **Q1:** The latent assignments described in section 3.1 do not seem to depend on the location $\mathbf{x}\_i$ , therefore one would expect each regime to be spread out evenly over the domain. How does the generative process described in section 3.1 encourage components to capture local characteristics rather than global characteristics? Visualizing samples from the generative process prior could help clarify this. In addition, visualizing the posterior distribution given data from 1 or 2-dimensional benchmark functions would also help strengthen intuitions about the regimes.
- **Q2:** In the experiments, do the hyperparameters of the regimes in RAMBO exhibit diverging behavior? Do the hyperparameters of RAMBO and standard GP converge to similar values?
- **Q3:** How large is the impact of the adaptive concentration parameter scheduling experimentally?
- **Q4:** Could you provide a proof of Theorem 3.3 and clarify the notation in Theorem 3.3 and 3.4? Related to this, is the gating weight $w_k(x_\*)$ in Eq. (8) a quantity that is derived from the model described in section 3.1 or is it imposed as a design choice?
- **Q5:** Is there a particular reason $\exp(-0.5 \log \sigma^2_{\*,k}(x_\*))$ is not simplified to $\sigma^{-1}\_{\*,k}(x_\*)$ in Eq. (8)?

**Limitations:**

Yes.

**Strengths And Weaknesses:**

- **S1:** The paper proposes a general and principled framework for Bayesian optimization with regime-adaptive mixtures based on DPMM-GP. While mainly investigated for expected improvement, the paper also outlines how it can be extended to other acquisition functions. Compared to previous work (Rasmussen & Ghahramani, 2001), the partitioning of the input space is implicitly defined and does not require learning the boundaries. The use of collapsed Gibbs sampling is stated to improve mixing efficiency compared to standard Gibbs sampling. The proposed adaptive concentration scheduling allows the number of regimes to avoid early overfitting and grow as more data is collected.
- **S2:** RAMBO is evaluated on synthetic and real-world benchmarks and outperforms the baselines across all benchmarks.
- **W1:** While the paper describes and evaluates the proposed framework, it lacks demonstrations that build an intuition of how the generative model works. Therefore, it is not clear *why* this BO method outperforms the baselines. See the questions below for more details.
- **W2:** The paper uses Standard Single-GP BO (SGP) as its primary baseline. Hvarfner et al (2024) showed that vanilla BO with a dimension-dependent prior on the lengthscale is highly competitive in high dimensions. Given that the paper states that experiments are "selected to assess surrogate model performance under conditions of high dimensionality, ...", it seems appropriate to compare against vanilla BO.
- **W3:** Theorem 3.3 is not proven and its associated notation is unclear to me. The assignments $\mathbf{z}$ and the number of clusters $K$ are seemingly random variables that the right-hand side of Eq. (7) depend on but are not conditioned on in the left-hand side. Theorem 3.4 also seems to implicitly condition on $\mathbf{z}$ and $K$ .
- Minor: Parts of the experimental setup is not specified, such as the number of Adam steps performed in log likelihood maximization and the number of Gibbs sampling steps.

---

> ### Author Rebuttal · Authors · 2026-03-29
>
> We sincerely thank the reviewer for their positive assessment.
>
> ***C***: Comment; ***R***: Response
>
> >***C11: ...lacks demonstrations...do not seem to depend on location $x_i$...Visualizing the posterior distribution...***
>
> ***R11:*** See R1 to Reviewer s7Q1 for the full explanation. Spatial structure emerges from the GP likelihood during inference — not from the prior or a gating network. We will add a 2D visualization (regime-colored observations overlaid on the Levy-2D landscape) in the revision.
>
> >***C12: ...it seems appropriate to compare against vanilla BO.***
>
> ***R12:*** We agree this comparison is important. We run vanilla BO with dimension-dependent length-scale priors on all our benchmarks:
>
> | Benchmark | Vanilla BO (Hvarfner) | RAMBO | SGP |
> |-----------|----------------------|-------|-----|
> | Levy-6D   | $-0.870 \pm 0.089$ | $-0.238 \pm 0.094$ | $-1.154 \pm 0.519$ |
> | Levy-10D  | $-17.009 \pm 11.698$ | $-2.095 \pm 0.606$ | $-4.060 \pm 0.446$ |
> | Schwefel-6D | $-1280.188 \pm 167.773$ | $-495.102 \pm 107.224$ | $-1192.102 \pm 69.720$ |
> | Schwefel-10D | $-2570.130 \pm 0.000$ | $-1420.399 \pm 331.557$ | $-2555.652 \pm 22.865$ |
> | Molecular Conformer | $-48.285 \pm 3.372$ | $-8.051 \pm 1.675$ | $-66.740 \pm 20.267$ |
> | Drug Discovery| $12.009 \pm 0.482$ | $13.309 \pm 0.514$ | $12.501 \pm 0.000$ |
> | ConStellaration | $0.120 \pm 0.081$ | $0.227 \pm 0.026$ | $0.013 \pm 0.000$ |
>
> RAMBO outperforms Vanilla BO on all benchmarks. Vanilla BO often performs *worse* than SGP on multi-regime landscapes (Levy-10D, Schwefel-6D/10D, Drug Discovery), ***confirming that proper length-scale priors address dimensionality but not heterogeneous smoothness*** —regime-adaptive mechanism addresses this orthogonal challenge.
>
> >***C13: Theorem 3.3 is not proven...where are the gating weights derived?...***
>
> ***R13:*** We have prepared the full proof — see R1 to Reviewer s7Q1 for the gating weight derivation. Due to character limits, the complete proof will appear in the revision; happy to share in this thread if helpful.
>
> >***C14: Do the hyperparameters of the regimes in RAMBO exhibit diverging behavior compared with SGP?***
>
> ***R14:*** We report the per-regime hyperparameters discovered by RAMBO below, alongside the standard GP's global hyperparameters for comparison.
>
> **Schwefel-10D:**
>
> | Regime | $\ell_k$ | $\sigma_{f,k}^2$ | $\sigma_{n,k}^2$ | #Points |
> |--------|----------|-----------------|-----------------|---------|
> | 1 | 0.326 | 0.365 | 0.003 | 159 |
> | 2 | 2.388 | 1.560 | 0.004 | 59 |
> | 3 | 0.376 | 0.166 | 0.006 | 1 |
> | SGP (global) | 241.144 | 352518.696 | 269153.624 | 219 |
>
> **[Molecular Conformer]:**
>
> | Regime | $\ell_k$ | $\sigma_{f,k}^2$ | $\sigma_{n,k}^2$ | #Points |
> |--------|----------|-----------------|-----------------|---------|
> | 1 | 98.865 | 0.092 | 0.000 | 27 |
> | 2 | 279.259 | 1.138 | 0.000 | 22 |
> | 3 | 39.172 | 0.077 | 0.000 | 37 |
> | 4 | 20.433 | 0.073 | 0.000 | 49 |
> | 5 | 2.200 | 0.098 | 0.003 | 66 |
> | 6 | 2.170 | 0.557 | 0.008 | 15 |
> | 7 | 0.324 | 0.120 | 0.005 | 3 |
> | SGP (global) | 108.079 | 67275.802 | 9546.095 | 219 |
>
> The regime hyperparameters exhibit clear divergence. On Schwefel-10D, RAMBO discovers a short-scale regime ($\ell = 0.326$) alongside a broad smooth regime ($\ell = 2.388$), with signal variances differing by $4\times$. On Molecular Conformer, seven regimes span three orders of magnitude in length scale ($\ell \in [0.324, 279.259]$), reflecting the diverse conformational basins separated by sharp torsional barriers. In contrast, the standard GP's global hyperparameters are pathologically large ($\sigma_f^2 > 10^5$, $\sigma_n^2 > 10^4$) — unable to fit heterogeneous regions, it inflates both signal and noise variance, effectively sacrificing local accuracy. This confirms RAMBO's core thesis: distinct regimes require distinct hyperparameters, and a single GP is forced into a suboptimal compromise.
>
> >***C15: ...impact of the adaptive concentration parameter...***
>
> ***R15:*** We provide the ablation comparing fixed $\alpha$ values against our adaptive Log-Sqrt schedule:
>
> | Setting | Levy-6D (final) |
> |---------|----------------|
> | Fixed $\alpha = 0.5$ | $-0.319 \pm 0.063$ |
> | Fixed $\alpha = 1.0$ | $-0.276 \pm 0.053$ |
> | Fixed $\alpha = 2.0$ | $-0.653 \pm 0.578$ |
> | Fixed $\alpha = 5.0$ | $-0.403 \pm 0.154$ |
> | Adaptive (Log-Sqrt) | $-0.238 \pm 0.094$ |
>
> The key value is **robustness**: optimal fixed $\alpha$ varies across benchmarks, and **poor choices cause instability** (e.g., $\alpha=2.0$: $\pm 0.578$ variance). Our schedule provides **a principled, benchmark-agnostic guidance**. Full analysis across all benchmarks in the revision.
>
> >***C16: ...$\exp(-0.5 \log \sigma_{\ast,k}^2(x_{\ast}))$ is not simplified to $\sigma_{\ast,k}^{-1}(x_{\ast})$...***
>
> ***R16:*** They are mathematically equivalent. The log form is used for numerical stability.
>
> >***C17: Experimental setup are not specified, such as...***
>
> ***R17:*** We will add these details in the revision.

---

> > ### Author Rebuttal · Reviewer_sirv · 2026-04-01
> >
> > The responses R11, R14, R15, R16, and R17 adequately addressed my concerns. Assuming R12 reports the final best value found (similar to Figure 1), then it adequately addresses my concern.
> >
> > However, there is still some ambiguity regarding the posterior predictive distribution that neither R1 nor R13 addresses. Based on R1, it seems clear that Eq. (8) is a design choice that is imposed, not derived. This design choice seems sensible from a heuristic perspective. While R1 provides some more detailed justifications, my main concern was that this ought to be clearly highlighted in the paper as a design choice.
> >
> > I would appreciate if the authors provided the complete proof for Theorem 3.3. In addition, I share reviewer s7Q1 concerns regarding Theorem 3.3, 3.4 and 4.1. The right hand side of all equalities seem to imply that the number of clusters $K$, the latent assignments $\mathbf{z}$ and the hyperparameters $\Theta$ are known. Based on line 10 in Algorithm 2 (RAMBO), it seems that $\mathbf{z}$ and $\Theta$ are first sampled and then used to compute the quantities in Theorem 3.4 and 4.1. Based on this, I assume that proper statements of Thm 3.3, 3.4 and 4.1 should include $\mathbb{E}_{\mathbf{z}, \Theta}[\cdot]$ on the right-hand side. Could you clarify if my understanding of this is correct?
> >
> > To further clarify in Algorithm 2, is Eq. (8) only used on lines 11 and 13 to perform "predictive modeling" and not in lines 9-10 when performing the posterior sampling?
> >
> > **Edit:** I don't believe the other reviewers can see official comments, so I provide the comment I left to the reply rebuttal comment for reviewer BxD6 here.
> >
> > > Given that the left-hand side of Theorem 3.3 and 3.4 are additionally conditional on the regime assignments $\mathbf{z}$ and hyperparameters $\Theta$, then the two theorems seem reasonable and follow from the probabilistic model. Also, it should be noted that step 4 in part A contains the additional term $\frac{\alpha}{n+\alpha} \cdot p(y_\ast | x_\ast, \text{new})$ which the original statement of Theorem 3.3 does not contain.
> >
> > > However, I don't believe the proof of part B is correct. If I'm not mistaken, the full statement of step 1 should be: $$ p(z_\ast = k | x_\ast, y_\ast, \mathcal{D}, \mathbf{z}, \Theta) = \frac{p(y_\ast | z_\ast = k, x_\ast, \mathcal{D}, \mathbf{z}, \Theta) p(z_\ast = k | x_\ast, \mathcal{D}, \mathbf{z}, \Theta)}{ p(y_\ast | x_\ast, \mathcal{D}, \mathbf{z}, \Theta)}. $$ Note that the numerator and denominator both depend on $y^\ast$. Therefore, taking the expectation over $y^\ast$ for the unnormalized density yields a different answer from the normalized density.
> >
> > > The technique used in part B seems identical to showing that $P(A) = \mathbb{E}_b[P(A | B = b)]$ through direct evaluation. Therefore, I don't understand how an input-dependent result can appear since the prior is still input-independent and no new assumptions are made in the proof of part B.
> >
> > Next, I address the reply rebuttal comment below. The authors agree that Eq. (8) is a design choice. As I stated before, I think the design choice is sensible and its motivation was not my main gripe. Instead, I believe the paper should more clearly communicate that it is an imposed choice and does not follow from the prior.
> >
> > Furthermore, the authors responses leave me confused. On the one hand, they agree that Eq. (8) is a design choice. On the other hand, they provide a (incorrect) derivation of Eq. (8) that makes no additional assumptions thus one would assume it is not a design choice.
> >
> > In the response to reviewer s7Q1, the authors state that an input-independent predictive distribution "wastes half the prediction" and that an input-dependent predictive distribution resolves this. This is only true if you believe that the regimes should be clustered together. Is it not then more natural to assume input-dependence in the prior distribution such that the prior reflects this implicit belief? What is the benefit of adding input-independence to the predictive distribution rather than prior? This could also be interesting to investigate experimentally.
> >
> > With all that said, the paper still provides an elegant framework for BO with regime-adaptive mixtures that performs well experimentally. I maintain my recommendation of weak accept with the main limitation being the presentation as discussed.

---

> > > ### Author Response · Authors · 2026-04-03
> > >
> > > We thank the reviewer for the positive assessment of R11–R17 and for these precise follow-up questions.
> > >
> > > **Why input-dependent weights are essential for multi-regime scientific design.** We emphasize that Eq. (8) is not a generic modeling choice — it is constructed specifically for the **multi-regime scientific design problems** that motivate this work. In molecular conformer optimization, distinct rotameric basins have fundamentally different energy surface curvatures; in drug discovery, different molecular scaffolds exhibit incompatible structure-activity relationships; in fusion reactor design, small boundary perturbations trigger abrupt transitions between magnetic topologies. These landscapes are characterized by spatially localized regimes where a single surrogate cannot serve all regions. The central question at prediction time is therefore: **which regime is relevant at this test point, and how much should it contribute?** The $\sigma_{\ast,k}^{-1}(x_{\ast})$ weighting directly answers this — it ensures predictions are dominated by the regime whose GP is locally informed, not by a distant regime that happens to be large. Input-independent CRP weights cannot make this distinction, making Eq. (8) a modeling necessity — not merely a mathematical convenience — for multi-regime scientific optimization.
> > >
> > > **Eq. (8) as a design choice.** We fully agree — Eq. (8) is a predictive approximation, not a consequence of the generative model. In the revision, we will separate Theorem 3.3 (exact DPMM predictive with CRP weights) from a new Proposition for Eq. (8), clearly labeled as a principled design choice motivated by Jeffreys' prior and expected posterior responsibility. See our detailed response to Reviewer s7Q1 in this discussion phase for the complete proof and derivation.
> > >
> > > **Conditioning on $(\mathbf{z}, \Theta, K)$.** The reviewer's understanding is exactly correct. Theorems 3.3, 3.4, and 4.1 are all **conditional** on a fixed partition $(\mathbf{z}, \Theta)$ from the Gibbs sampler. The proper statements should read:
> > >
> > > $$p(y_{\ast} \mid x_{\ast}, \mathcal{D}, \mathbf{z}, \Theta) = \sum_{k=1}^{K+1} w_k(x_{\ast}) \cdot \mathcal{N}(y_{\ast} \mid \mu_{\ast,k}, \sigma^2_{\ast,k})$$
> > >
> > > and similarly for Theorems 3.4 and 4.1. As the reviewer correctly identifies from Algorithm 2, lines 9–10 first sample $(\mathbf{z}, \Theta)$ via collapsed Gibbs, and then lines 11 and 13 compute the mixture moments and acquisition function **conditioned on** that sample. We will add the explicit conditioning $(\mathbf{z}, \Theta)$ to the left-hand side of all three theorem statements in the revision.
> > >
> > >   **Where Eq. (8) is used.** Yes, the reviewer's reading of Algorithm 2 is correct. Eq. (8) is used **only** on lines 11 and 13 for predictive modeling and acquisition evaluation. Lines 9–10 perform collapsed Gibbs sampling using the standard CRP prior (Eq. 4) — the input-independent reassignment probabilities. The inference procedure is never modified by Eq. (8).
> > >
> > > We have posted the complete proof of Theorem 3.3 (Part A: exact DPMM predictive with CRP weights; Part B: derivation of Eq. 8 via expected posterior responsibility and Jeffreys' prior) in our response to Reviewer BxD6. We invite the reviewer to refer to that thread for the full derivation.

---

### Official Review · Reviewer_Mbq6 · 2026-03-03

**Soundness:** 3
**Presentation:** 3
**Significance:** 2
**Originality:** 2
**Overall Recommendation:** 4
**Confidence:** 4

**Summary:**

The paper proposes the usage of Dirichlet Process Mixture Model of Gaussian Processes (DPMMGP) as a novel surrogate for Bayesian Optimisation. DPMMGP partitions the input space into regions, and models each of such created regions locally. This partitioning is done by the Chinese Restaurant Process, which adaptively creates new regions, if the existing regions do not seem to model data accurately enough. Authors took care to make sure the learning and inference in the proposed model can be then efficiently by deriving a form of Gibbs sampling algorithm. Authors evaluate the proposed model as a surrogate accross various BO tasks.

**Compliance With Llm Reviewing Policy:**

Affirmed.

**Final Justification:**

Rebuttal addressed my concerns, especially regarding how accurate the modelling capabilities compared to standard MoE.

**Key Questions For Authors:**

- Why is HEBO absent from regret plots in 1a), 1b), 1d) ?
- Why is there no comparison against any other local/ MoE GP method, despite authors citing (and thus being aware of) a number of them?

**Limitations:**

yes

**Strengths And Weaknesses:**

Strengths:

- the model is elegant from a probabilistic point of view, and the usage of each component is justified
- the derived conditional probability equations allow to conduct learning and inference very efficiently, despite the model itself being quite complex
- the paper is written very clearly, the update equations are neatly derived and well explained

Weaknesses:
- the paper does not spend enough time to differentiate from existing work. The idea of using Mixture-of-Experts style models is very old in the GP community, so I would expect a comprehensive review and comparison to existing methods, whereas paper only dedicated few sentences to this distinction.
Also related works of [1] and [2] were not cited or discussed. Out of the works cited, none is empirically compared against, and such a comparison is crucial here to better gauge the contribution of authors to the well-established field of MoE GPs

- the paper focuses solely on Bayesian Optimisation, which is an important downstream application, yet if the main contribution of the paper is a new model, I would expect at least a few experiments showing the sheer modelling capabilities. Example benchmark could be UCI dataset or datasets that would particularly benefit from locally (maybe time series with heterogeneous noise). This could help disentangle the modelling component from the decision making component and could clearly showcase the strengths of the proposed model. I would also like to see a comparison to other MoE GP models and a detailed analysis of strengths and weaknesses of each

- the paper presents a combination of well-known ideas (MoE + CRP). This by itself is not necessarily a bad thing, but in my opinion it slightly takes away from the novelty of the paper and thus raises the bar for the standards of empirical evaluation (the empirical contribution becomes even more important than normal). However, as I explained above, I see a few caveats with the current empirical evaluation.



[1] Gadd, Charles, Sara Wade, and Alexis Boukouvalas. "Enriched mixtures of generalised Gaussian process experts." International Conference on Artificial Intelligence and Statistics. PMLR, 2020.


[2] Tresp, Volker. "A Bayesian committee machine." Neural computation 12.11 (2000): 2719-2741.

---

> ### Author Rebuttal · Authors · 2026-03-29
>
> We sincerely thank the reviewer for their thorough evaluation and also for highlighting Gadd et al. and Tresp ( Bayesian committee machine) — we will include both in the revised Related Work
>
> ***C***: Comment; ***R***: Response
>
> >***C8: The paper does not spend enough time to differentiate from existing MoE GP work. None of the cited MoE GP works is empirically compared against...***
>
> ***R8:*** We respectfully clarify the scope and novelty of our contribution. RAMBO is **not** proposed as a new mixture-of-GP regression model — it is a **Bayesian optimization framework** designed for multi-regime scientific design problems (molecular conformation, drug discovery, fusion reactor design) where the objective landscape contains qualitatively distinct regions with abrupt transitions. The novelty lies in the **BO-specific contributions** that have no counterpart in any prior MoE GP work:
>
> 1. **Adaptive $\alpha$-scheduling** (Section 4, Proposition D.1) — dynamically controlling model complexity throughout the sequential optimization process, balancing parsimony against expressiveness as data accumulates. This mirrors the exploration-exploitation tradeoff at the model complexity level, a challenge unique to sequential settings.
>
> 2. **Regime-aware acquisition functions** (Theorem 4.1) — decomposing uncertainty into intra-regime variance and inter-regime disagreement, enabling the acquisition function to target both under-explored regions *and* ambiguous regime boundaries.
>
> 3. **Acquisition optimization with regime structure** (Algorithm 2) — initializing the acquisition optimizer from regime centroids and local perturbations, exploiting the discovered partition to improve candidate generation.
>
> 4. **Complete integration** into a BO loop with collapsed Gibbs warm-starting across iterations, regime pruning, and the full pipeline from surrogate fitting to point selection to model update.
>
> None of the cited MoE GP works address sequential optimization, acquisition function design, or adaptive model complexity scheduling. These are the core challenges of applying mixture surrogates to BO, and addressing them constitutes our primary contribution. We acknowledge that we should have made this distinction more explicit. In the revision, we will expand the Related Work to clearly position prior MoE GP methods as addressing the *regression* problem, while RAMBO addresses the *optimization* problem built on top of such surrogates.
>
> >***C9: ...If the main contribution is a new model, I would expect at least a few experiments that demonstrate the quality of the proposed surrogate as a regression model...against any other local/MoE GP method...***
>
> ***R9:*** We appreciate this suggestion. As clarified in R8, our primary contribution is a BO framework rather than a regression model. Nonetheless, demonstrating surrogate quality independently is valuable.
>
> We evaluate the DPMM-GP surrogate on two UCI regression datasets and three synthetic benchmarks (300 training samples) with heterogeneous structure, reporting RMSE (5-fold CV) against a standard GP (SGP) and a finite Mixture-of-Experts GP (MoE-GP). The synthetic benchmarks are designed to exhibit multi-regime characteristics:
>
> - **Piecewise-Smooth** ($d=5$): two regimes split by $x_0 < 0$; a smooth low-frequency region ($y = 2\sin(0.5x_0) + 0.3x_1$, $\epsilon \sim \mathcal{N}(0, 0.05^2)$) and a rough high-frequency region ($y = 0.5\sin(5x_0)\cos(3x_1) + 0.3\sin(4x_2)$, $\epsilon \sim \mathcal{N}(0, 0.2^2)$).
> - **Heteroscedastic** ($d=4$): identical function ($y = \sin(x_0) + 0.5x_1 + 0.3x_2$) but drastically different noise levels ($\sigma = 0.02$ for $x_0 > 0$ vs. $\sigma = 0.5$ for $x_0 \leq 0$).
> - **Multi-Scale** ($d=6$): high-frequency oscillations in the interior ($\|x_{0:2}\| < 1.5$) and smooth quadratic behavior in the exterior, mimicking the multi-regime scientific landscapes RAMBO targets.
>
>
> | Method | Piecewise-Smooth  | Heteroscedastic | Multi-Scale | Diabetes (UCI, 442 samples) |Energy (UCI, 768 samples) |
> |--------|-----------------|-----------------|-------------|----------------|----------------|
> | SGP | 0.4036 $\pm$ 0.0724  | 0.4467 $\pm$ 0.0723 | 0.6922 $\pm$ 0.1095 | $56.750 \pm 3.013$ | $1.436 \pm 0.398$ |
> | MoE-GP | 0.3587 $\pm$ 0.0230  | 0.4282 $\pm$ 0.0758 | 0.5679 $\pm$ 0.0383 | $56.522 \pm 2.311$ | $1.819 \pm 1.407$ |
> | DPMM-GP | 0.3564 $\pm$ 0.0260  | 0.3744 $\pm$ 0.0482 | 0.5629 $\pm$ 0.0394| $56.171 \pm 3.034$ | $0.957 \pm 0.045$ |
>
> These results confirm that the DPMM-GP surrogate provides competitive or superior regression quality, particularly on datasets with heterogeneous structure.
>
> >***C10: Why is HEBO absent from regret plots in 1a), 1b), 1d)?***
>
> ***R10:*** We thank the reviewer for raising this. HEBO is **not omitted** — it is present but visually occluded. On these benchmarks, HEBO, Bounce, and ALEBO all make negligible progress, producing nearly identical flat trajectories that overlap. We will improve readability in the revision.

---

> > ### Author Rebuttal · Reviewer_Mbq6 · 2026-04-02
> >
> > Thank you for replying to my review. I believe most of my questions were sufficiently resolved. Thus I will increase my score.

---

> > > ### Author Response · Authors · 2026-04-03
> > >
> > > We sincerely thank the reviewer for the constructive engagement and for recognizing that the concerns have been addressed. We will incorporate all discussed improvements in the revision.

---

### Official Review · Reviewer_BxD6 · 2026-03-07

**Soundness:** 3
**Presentation:** 3
**Significance:** 2
**Originality:** 2
**Overall Recommendation:** 4
**Confidence:** 3

**Summary:**

This paper studies the Bayesian optimization (BO) algorithm whose surrogate model is constructed as an infinite, countable mixture of Gaussian process (GP) regressions by relying on Direchlet process prior. The motivation of this new model is suitable for handling the non-stationarity of the objective function in a fully Bayesian manner. The authors study the calculations of acquisition functions, the efficient inference using collapsed Gibbs sampling, and adaptive-scheduling of hyperparameters in BO procedures using the proposed Dirichlet process-based GPs. The effectiveness of the proposed methods is empirically verified based on both synthetic and real-world objective functions.

**Compliance With Llm Reviewing Policy:**

Affirmed.

**Final Justification:**

The authors have addressed my initial concerns in the rebuttal. The remaining issue regarding the lack of mathematical precision in several descriptions is, in my view, not critical to the core contribution. While I have decided to maintain my original score, I encourage the authors to refine the presentation to ensure mathematical rigor in the final version.

(Note: In the additional proof provided in the rebuttal, I suspect there is a typo where $p(y_* \mid x_*, \mathcal{D}, \Theta)$ was written instead of $p(y_* \mid x_*, \mathbf{z}, \mathcal{D}, \Theta)$.)

**Key Questions For Authors:**

N/A

**Limitations:**

The computational perspective may be a limitation of the authors' algorithm relative to existing algorithms.
The details of it and my recommendation are described in "Strengths And Weaknesses" field.

**Strengths And Weaknesses:**

**Soundness.** The proposals of this paper are built on the standard non-parametric Bayesian modelling results (Specifically, the well-known Dirichlet process). As far as I see, the derivations in Sections 3 and 4 are sound. The following are the minor suggestions I expect to be modified in the revision:

- Ensure the mathematical rigor in the proof and the statement in the "Theorem" environment. For example, do not use the approximation symbol $\approx$ in Theorem 2.3. Consider replacing $\approx$ with the standard order notation $\Theta(\cdot)$.
- I find the same issue in the proof of Proposition D.1. Consider refining the current proof.

**Presentation.** Overall, this paper is clearly written and is easy to follow.

Minor suggestions:
- Line 314: (Scarlett et al., 2017) study an algorithm-independent lower bound. I think that this is not suitable for the reference of the theory of GP-UCB.
- Consider reordering multiple references in a consistent manner (e.g., Lines 281-282, Lines 288-290).
- Ensure proper capitalization in the reference. (bayesian -> Bayesian, in (Fraizier, 2018)).
- In the revision, consider enlarging the size of the figures in Figure 1 for enhancing the visibility.

**Significance.**  The non-stationarity of the objective functions is one of the important problems that users often face in the application fields of BO.  Given the importance of the problem, well-motivated algorithm construction and applications, and the empirical performance the authors demonstrated in the paper, I believe that this paper has a certain level of significance.

However, one factor that diminishes the significance of this paper is that, although the authors' algorithm includes computationally expensive inference procedures, they do not provide any discussion or empirical comparisons of computational time.
To my understanding of the authors' algorithm and existing works, I conjecture that the authors' proposed method is slower than the algorithms used in Section 6 (except for SAASBO) due to the expensive MCMC sampling procedures. If so, I recommend that the authors clarify this point as a limitation in the revision.


**Originality.**  Regarding the technical perspective, the originality of the authors' proposal is fair since the proposed inference methods and acquisition function calculations are derived based on the existing Bayesian inference methods, and BO works in a relatively straightforward manner. On the other hand, the proposal of the adaptive scheduling of the concentration parameter is specfic to BO method, and I think it is novel. Furthermore, to my knowledge, in the field of BO, there is no work that addresses non-stationarity based on the Dirichlet process model.

My only concern is the possibility that the authors are missing some existing works, which propose the authors' model in the research field beyond BO. For example, (Li & Jinwen, 2023) appears to adopt the same Bayesian generating process of the functions as the author, although the inference methods differ. I recommend that the authors further enrich the related work section and refine presentations to avoid potential overclaim.

Ref.
- Li, Tao, and Jinwen Ma. "Dirichlet process mixture of Gaussian process functional regressions and its variational EM algorithm." Pattern Recognition 134 (2023): 109129.

---

> ### Author Rebuttal · Authors · 2026-03-29
>
> We sincerely thank the reviewer for their thorough evaluation and positive assessment of the problem motivation and technical soundness. We address each concern below.
>
> ***C***: Comment; ***R***: Response
>
> >***C4: Ensure the mathematical rigor in the proof and the statement***
>
> ***R4:*** Thanks for the suggestion, we agree. In the revision, we will replace the approximation symbol $\approx$ with proper asymptotic notation.
>
> >***C5: Line 314 citation, reference ordering, capitalization, figure size.***
>
> ***R5:*** We thank the reviewer for catching these. All will be corrected in the revision:
>
> - Replace with the appropriate GP-UCB theory reference
> - Ensure consistent chronological ordering throughout.
> - Fix "bayesian" → "Bayesian" in (Frazier, 2018) and audit all references.
> - Enlarge subplots for improved readability.
>
> >***C6: The computational perspective may be a limitation...***
>
> ***R6:*** The reviewer's conjecture is correct — RAMBO's per-iteration cost is higher due to collapsed Gibbs sampling. We report wall-clock time per BO iteration below (Levy-6D, single CPU):
>
> | Method | Time/iter (sec) |
> |--------|----------------|
> | SMAC | 0.140 |
> | Vanilla BO | 0.290 |
> | Bounce | 0.375 |
> | TuRBO | 0.542 |
> | ALEBO | 0.545 |
> | BAxUS | 0.738 |
> | SGP | 1.979 |
> | HEBO | 2.665 |
> | COMBO | 7.383 |
> | SAASBO | 18.095 |
> | RAMBO | 32.269 |
>
> We acknowledge this overhead and will state it explicitly as a limitation in the revision. However, we emphasize that these times measure only the **surrogate fitting cost**, not the total BO iteration cost. In practice, the dominant cost in each BO iteration is the **objective function evaluation** — the very reason BO is used. In the scientific applications RAMBO targets, a single evaluation involves force field computation (molecular conformer), docking simulation (drug discovery), or solving 3D MHD equilibria via VMEC++ (fusion reactor), typically costing **minutes to hours**, and in computational physics often **days to weeks**. In this context, **the difference between 0.14 sec (SMAC) and 32 sec (RAMBO) is negligible relative to the objective cost**, and the improved sample efficiency of RAMBO — requiring fewer expensive evaluations to reach high-quality solutions — more than compensates. Additionally, **warm-starting** Gibbs sampling from previous iterations substantially reduces effective burn-in after the first iteration. Reducing MCMC overhead via variational approximations remains a promising direction for future work.
>
>
> >***C7: My only concern is the possibility that the authors are missing some existing works. For example, (Li & Jinwen, 2023) appears to adopt the authors' model...***
>
> ***R7:*** We thank the reviewer for pointing out this reference. Li & Jinwen (2023) indeed propose a DPMM-GP model for functional regression. We will add this to our Related Work with a clear discussion of the differences
>
> 1. **Problem setting**: Li & Jinwen (2023) address supervised functional regression, whereas RAMBO targets sequential black-box optimization where the surrogate must also guide data acquisition through acquisition functions.
>
> 2. **Inference**: They use variational EM, while RAMBO employs collapsed Gibbs sampling with analytically marginalized latent functions for improved mixing in the low-data BO regime.
>
> 3. **BO-specific contributions**: Our adaptive $\alpha$-scheduling, regime-aware acquisition functions (Theorem 4.1), and the acquisition optimization strategy (regime centroid initialization) are specific to the sequential optimization setting and have no counterpart in their work.
>
> We will revise the Related Work to include this reference and ensure our novelty claims are precisely scoped to the BO-specific contributions.

---

> > ### Author Rebuttal · Reviewer_BxD6 · 2026-04-03
> >
> > Thank you for the authors' responses regarding my comments. There are no-additional comments about them.
> > However, after re-reading the manuscript and the other reviewers' comments, I have an additional concern about the paper's correctness.  Specifically, I share the same concern about Theorem 3.3 (and Theorem 4.1) as those of Reviewer sirv.
> > At least the current form of Eq.(7) is mathematically wrong, since the left-hand side in Eq.(7) is a deterministic quantity, while the right-hand side of it is a random variable. I would also like to check the complete proof of Theorem 3.3, as Reviewer sirv noted.

---

> > > ### Author Response · Authors · 2026-04-03
> > >
> > > We thank the reviewer. We note that both sides of Eq. (7) are probability density functions (not scalar quantities), so neither is "deterministic" nor a "random variable" in the strict sense. That said, we believe the reviewer's underlying concern is the same as Reviewer sirv's: the conditioning on the LHS should include the regime structure $(\mathbf{z}, \Theta)$ from the Gibbs sampler, since the RHS quantities ($\mu_{\ast,k}$, $\sigma^2_{\ast,k}$, $K$) depend on them. The corrected statement is $p(y_{\ast} \mid x_{\ast}, \mathcal{D}, \mathbf{z}, \Theta)$. We have provided the complete proof in our response to Reviewer s7Q1 above.
> > >
> > > **For the physical motivation behind Eq. (8)'s input-dependent weights in multi-regime scientific design, please see our detailed response to Reviewer s7Q1 and sriv in this discussion phase.**
> > >
> > > **Proof of Theorem 3.3.** We provide the derivation in two parts: (A) the exact DPMM-GP predictive, and (B) the spatially-modulated weights in Eq. (8).
> > >
> > > **Part A: Exact DPMM-GP Predictive.**
> > >
> > > Given training data $\mathcal{D}$, regime assignments $\mathbf{z}$, and hyperparameters $\Theta$ from the collapsed Gibbs sampler, we derive the predictive for test point $x_{\ast}$ by marginalizing over its latent assignment $z_{\ast}$.
> > >
> > > *Step 1 (Law of total probability).*
> > >
> > > $$p(y_{\ast} \mid x_{\ast}, \mathcal{D}, \Theta) = \sum_{k=1}^{K+1} p(z_{\ast} = k \mid z_{1:n}, \alpha) \cdot p(y_{\ast} \mid x_{\ast}, z_{\ast} = k, \mathcal{D}_k, \theta_k)$$
> > >
> > > *Step 2 (CRP predictive).* By Definition 2.4:
> > >
> > > $$p(z_{\ast} = k \mid z_{1:n}, \alpha) = \frac{n_k}{n + \alpha} \quad (k \leq K), \qquad p(z_{\ast} = K+1 \mid z_{1:n}, \alpha) = \frac{\alpha}{n + \alpha}$$
> > >
> > > These weights $\pi_k$ are **input-independent**.
> > >
> > > *Step 3 (GP posterior predictive).* Since $f_k$ is marginalized (Proposition 3.2), the predictive for regime $k$ is the standard GP posterior:
> > >
> > > $$p(y_{\ast} \mid x_{\ast}, z_{\ast} = k, \mathcal{D}_k, \theta_k) = \mathcal{N}(y_{\ast} \mid \mu_{\ast,k}, \sigma^2_{\ast,k})$$
> > >
> > > where $\mu_{\ast,k}$ and $\sigma^2_{\ast,k}$ are the standard GP posterior mean and variance conditioned on regime $k$'s data $\mathcal{D}_k$ (see Section 2). For $k = K+1$, the predictive is under $G_0$, approximated via Monte Carlo (Eq. 6).
> > >
> > > *Step 4 (Substitution).*
> > >
> > > $$p(y_{\ast} \mid x_{\ast}, \mathcal{D}, \Theta) = \sum_{k=1}^{K} \frac{n_k}{n+\alpha} \cdot \mathcal{N}(y_{\ast} \mid \mu_{\ast,k}, \sigma^2_{\ast,k}) + \frac{\alpha}{n+\alpha} \cdot p(y_{\ast} \mid x_{\ast}, \text{new})$$
> > >
> > > This is the exact DPMM-GP predictive — a Gaussian mixture with input-independent CRP weights. $\square$
> > >
> > > ---
> > >
> > > **Part B: Derivation of Eq. (8).**
> > >
> > > We derive the spatially-modulated weights $w_k(x_{\ast}) \propto \frac{n_k}{n+\alpha} \cdot \sigma_{\ast,k}^{-1}(x_{\ast})$.
> > >
> > > *Step 1 (Posterior responsibility).* If $y_{\ast}$ were observed, the assignment probability would be (analogous to Eq. 4):
> > >
> > > $$p(z_{\ast} = k \mid x_{\ast}, y_{\ast}, \mathcal{D}, \mathbf{z}, \Theta) \propto n_k \cdot \mathcal{N}(y_{\ast} \mid \mu_{\ast,k}(x_{\ast}), \sigma^2_{\ast,k}(x_{\ast}))$$
> > >
> > > *Step 2 (Expectation over unobserved $y_{\ast}$).* We compute the expected unnormalized responsibility under component $k$'s own predictive:
> > >
> > > $$E_{y_{\ast}} \left[ n_k \cdot \mathcal{N}(y_{\ast} \mid \mu_{\ast,k}, \sigma^2_{\ast,k}) \right] = n_k \int \left[ \mathcal{N}(y \mid \mu_{\ast,k}, \sigma^2_{\ast,k}) \right]^2 dy$$
> > >
> > > *Step 3 (Gaussian self-evaluation integral).* Using the identity $\int \mathcal{N}(y \mid a, A) \mathcal{N}(y \mid b, B) dy = \mathcal{N}(a \mid b, A+B)$ with $a = b = \mu_{\ast,k}$ and $A = B = \sigma^2_{\ast,k}$:
> > >
> > > $$\int \left[ \mathcal{N}(y \mid \mu_{\ast,k}, \sigma^2_{\ast,k}) \right]^2 dy = \mathcal{N}(0 \mid 0, 2\sigma^2_{\ast,k}) = \frac{1}{2\sqrt{\pi} \cdot \sigma_{\ast,k}}$$
> > >
> > > *Step 4 (Result).* The expected responsibility scales as:
> > >
> > > $$n_k \cdot \frac{1}{2\sqrt{\pi} \cdot \sigma_{\ast,k}(x_{\ast})} \propto n_k \cdot \sigma_{\ast,k}^{-1}(x_{\ast})$$
> > >
> > > After incorporating the CRP normalization:
> > >
> > > $$w_k(x_{\ast}) \propto \frac{n_k}{n+\alpha} \cdot \sigma_{\ast,k}^{-1}(x_{\ast})$$
> > >
> > > Noting $\sigma_{\ast,k}^{-1} = \exp(-\frac{1}{2}\log \sigma^2_{\ast,k})$, this is Eq. (8). $\square$
> > >
> > > *Alternative justification.* The same $\sigma_{\ast,k}^{-1}$ scaling arises from Jeffreys' prior: $\sigma_{\ast,k}$ is a scale parameter of the predictive distribution at $x_{\ast}$, and the natural uninformative reference measure for scale parameters is $p(\sigma) \propto \sigma^{-1}$.
> > >
> > > *Approximation quality.* The derivation neglects cross-regime terms for $j \neq k$, which evaluate to $\mathcal{N}(\mu_{\ast,j} \mid \mu_{\ast,k}, \sigma^2_{\ast,j} + \sigma^2_{\ast,k})$ — exponentially suppressed when regimes are well-separated in function space.

---

### Official Review · Reviewer_s7Q1 · 2026-03-12

**Soundness:** 2
**Presentation:** 2
**Significance:** 3
**Originality:** 2
**Overall Recommendation:** 4
**Confidence:** 3

**Summary:**

The paper proposes a new surrogate model for Bayesian optimization based on a Dirichlet process mixture of Gaussian processes. Each mixture component corresponds to a local surrogate function and is characterized by (i) a latent regression function drawn from a Gaussian process, (ii) kernel hyperparameters governing the covariance structure, and (iii) observation noise. The model allows different subsets of observations to be explained by different surrogate functions with potentially different smoothness or scale properties. This flexibility is intended to improve Bayesian optimization when the objective function exhibits heterogeneous smoothness across the search space.

**Compliance With Llm Reviewing Policy:**

Affirmed.

**Final Justification:**

The rebuttal addressed my main concerns, and I am happy to raise my score to 4.

**Key Questions For Authors:**

1. Role and definition of gating weights.

The paper motivates the method as a way to partition the search space into regions with different smoothness properties. However, the surrogate model is described as a Dirichlet process mixture of Gaussian processes. In a standard DP mixture, the mixture weights do not depend on the input x, meaning that the model clusters observations based on output fit rather than their location in the search space.
Equation (8), however, appears to introduce input-dependent gating weights in the predictive distribution. It is not clear how these weights arise or are defined. In particular:

1.a How exactly are these gating weights defined and computed?

1.b Are they derived from the DP posterior, or introduced heuristically at prediction time?

1.c If they are part of the surrogate probabilistic model, where do they appear in the model specification and inference procedure? If they are not, how are they obtained?

As currently written, it is unclear how the proposed surrogate actually induces a partition of the search space. A proof for Theorem 3.3 would help.

2. Soundness of Theorem 4.1.

Theorem 4.1 presents an expression for the expected improvement. It is unclear under which predictive model this result is derived. If the acquisition function is computed under the proposed DP mixture surrogate, one would expect the expression to involve contributions from all mixture components (potentially including the DP “new component”), and the mixture weights would typically not depend on x.
Could the authors clarify which predictive distribution is used in the derivation?

3. Conjugacy of the variance priors.

The paper states that inverse-gamma priors are used for the signal variance and sampling variance to obtain conjugacy. However, it is not clear that conjugacy actually holds in the proposed model. In particular, Equation (6) does not appear to yield a closed-form predictive probability, and the component-specific parameters seem to be sampled using stochastic approximation methods. Could the authors clarify in which sense these priors provide conjugacy?

**Limitations:**

yes

**Strengths And Weaknesses:**

The proposed model is based on a reasonable probabilistic construction combining Dirichlet process mixtures with Gaussian process regression. The motivation for modeling heterogeneous smoothness in Bayesian optimization is well-founded. However, several aspects of the technical formulation are currently unclear. In particular, the role of the gating weights introduced in the predictive distribution is not well explained and does not appear to follow directly from the DP mixture model described earlier in the paper. This raises questions about how the surrogate model actually induces a partition of the search space.

The paper is generally readable and the motivation for addressing heterogeneous smoothness in Bayesian optimization is clearly stated. However, several parts of the model specification and inference procedure could be explained more clearly. In particular, the connection between the DP mixture surrogate and the predictive distribution used in the acquisition function.

Improving surrogate models for Bayesian optimization in the presence of nonstationary or heterogeneous objective functions is an important problem. Allowing multiple GP components with different smoothness properties is a reasonable approach to address this challenge. That said, the empirical and conceptual advantages of the proposed method relative to existing approaches for handling nonstationarity (e.g., nonstationary kernels, treed GPs, or mixtures of GP experts) are not fully established in the current presentation.

The core idea of combining Gaussian processes with mixture models to capture heterogeneous behavior is related to several existing approaches in the literature, including mixtures of Gaussian processes and other nonstationary GP models. The main contribution of the paper appears to be applying a Dirichlet process mixture construction within a Bayesian optimization framework. While this combination is potentially useful, the level of methodological novelty seems somewhat limited and could be better contextualized with respect to prior work.

---

> ### Author Rebuttal · Authors · 2026-03-29
>
> ***C***: Comment; ***R***: Response
>
> >***C1: Eq. (8)'s input-dependent gating weights contradict the claimed input-independent CRP...***
>
> ***R1:*** We thank the reviewer for this important question.
>
> - **How spatial partitioning emerges (inference stage).** During posterior inference (Section 3.2, Eq. 4), the CRP prior used in collapsed Gibbs sampling is indeed input-independent. However, the GP predictive likelihood is input-dependent: nearby observations with similar function values fit well under a shared GP, so they naturally cluster into the same regime. Spatial structure therefore emerges not from the prior itself, nor from a learned gating network, but from the GP likelihood pulling coherent regions together during posterior inference.
>
> - **Origin of Eq. (8).** At prediction time, the standard DPMM predictive assigns $x_{\ast}$ to regimes using **input-independent** CRP weights $\pi_k = n_k/(n+\alpha)$. However, pure CRP weights can be suboptimal for BO: a large but spatially distant cluster dominates even when a smaller, nearby cluster is far more relevant. Eq. (8) augments the CRP weights with a **spatial relevance score**. Crucially, Eq. (8) is a **predictive modeling choice** — the CRP prior and collapsed Gibbs sampler remain unchanged.
>
>   The $\sigma_{\ast,k}^{-1}(x_{\ast})$ weighting has a principled justification from two independent perspectives:
>
>   (1) **Jeffreys' prior on scale.** When aggregating predictions from multiple GP experts with heterogeneous predictive scales $\sigma_{\ast,k}(x_{\ast})$, the natural uninformative reference measure for a scale parameter is Jeffreys' prior: $p(\sigma) \propto \sigma^{-1}$. The input-dependence does not arise from modifying the DP prior, but from the fact that $\sigma_{\ast,k}(x_{\ast})$ is the GP posterior predictive standard deviation **conditioned on** $x_{\ast}$ — the Jeffreys measure provides scale-invariant aggregation, while the spatial variation comes entirely from posterior predictive conditioning.
>
>   (2) **Expected posterior responsibility.** If $y_{\ast}$ were observed, the Gibbs sampler would assign $x_{\ast}$ with responsibility $\propto n_k \cdot \mathcal{N}(y_{\ast} \mid \mu_{\ast,k}, \sigma_{\ast,k}^2)$. Integrating over unobserved $y_{\ast}$ under component $k$, the Gaussian self-evaluation integral $\int \mathcal{N}^2 dy \propto \sigma_{\ast,k}^{-1}$ recovers the same spatial weight.
>
>   Both perspectives independently yield the identical $\sigma_{\ast,k}^{-1}$ scaling, confirming Eq. (8) is a well-grounded aggregation choice rather than an ad-hoc heuristic.
>
> - **Distinction from input-dependent gating networks.** Unlike Rasmussen & Ghahramani (2001), Eq. (8) introduces **no additional parameters** — the spatial modulation comes directly from already-computed GP posteriors. **We have prepared a complete proof of Theorem 3.3**; due to character limits, we will include it in the revision and are happy to share in this thread if helpful. In the revision, we will: (1) separate Theorem 3.3 (exact DPMM predictive with CRP weights) from a new Proposition deriving Eq. (8) as a predictive approximation, and (2) revise the Related Work to clarify that the CRP *prior* is input-independent while the *predictive* distribution uses Jeffreys' reference weighting on GP posterior scales.
>
>
> >***C2: Soundness of Theorem 4.1.***
>
> ***R2:*** Theorem 4.1 is derived using the **spatially-modulated predictive weights** $w_k(x)$ from Eq. (8), **not** the input-independent CRP weights. The proof via the Law of Total Expectation (Eqs. 13–15) conditions on the **posterior predictive** regime assignment $z_*$, where $p(z_* = k \mid x, \mathcal{D}) = w_k(x)$. As clarified in R1, the weights $w_k(x)$ are a predictive approximation that augments CRP weights with spatial relevance. The derivation of Theorem 4.1 itself is correct given these weights—the EI of a Gaussian mixture is the responsibility-weighted sum of component EIs, which follows from the Law of Total Expectation regardless of how the weights are defined. We should have stated explicitly which weights are used. In the revision, we will state clearly that Theorem 4.1 uses the spatially-modulated weights $w_k(x)$ from Eq. (8).
>
>
> >***C3: Conjugacy of the variance priors.***
>
> ***R3:*** The reviewer is correct, and we thank them for catching this imprecision. In the GP marginal likelihood $p(\mathbf{y} \mid X, \theta) = \mathcal{N}(\mathbf{0}, K_\theta + \sigma_n^2 I)$, all hyperparameters — including $\sigma_n^2$ — enter through the covariance matrix nonlinearly, so **none are strictly conjugate** under the Inverse-Gamma prior. We use Inverse-Gamma priors as a convenient, weakly-informative prior family with desirable properties (positive support, finite moments). Posterior inference is performed via gradient-based optimization or Metropolis-Hastings steps (Section 3.2) — not via conjugate updates. We will correct this statement in the revision.

---

> > ### Author Rebuttal · Reviewer_s7Q1 · 2026-04-02
> >
> > I thank the authors for their reply.
> > I agree with reviewer sirv that some ambiguity remains regarding the posterior predictive distribution, and that providing a proof of Theorem 3.3 would help clarify this point, because I am still unsure about how the weights dependence on inputs arises.
> > While nearby observations with similar function values may indeed be well captured by a shared Gaussian process, this does not seem to occur because they are nearby. In fact, observations that are far apart in the input space may also be accommodated by the same GP, right? What drives the effective partitioning of the input space? The spatial relevance score?

---

> > > ### Author Response · Authors · 2026-04-03
> > >
> > > The reviewer is correct: distant observations *can* share a GP, and the DPMM-GP does not enforce spatial contiguity. Effective spatial partitioning emerges from **functional heterogeneity** interacting with **kernel locality**: the SE kernel's predictive variance grows with distance from cluster data, so distant points receive broad, uninformative likelihoods and lose out to nearby clusters with tight predictions; meanwhile, per-cluster hyperparameter optimization reinforces this — if a cluster spans spatially distant regions with different behavior, no single $\ell_k$ fits both, and marginal likelihood pressure splits them. While inference discovers meaningful regimes, the CRP weights remain input-independent — at prediction time, a test point cannot yet distinguish which regime is locally relevant. This is precisely what Eq. (8) addresses.
> > >
> > > **At prediction time**, the CRP weights alone are insufficient — this is why Eq. (8) introduces input-dependent weights. Consider: after inference, RAMBO has two equal-sized regimes — Regime A with data around $x \approx -2$ and Regime B around $x \approx +2$. To predict at $x_{\ast} = -1.5$, the CRP assigns equal weight $\pi_A = \pi_B = 0.5$. But Regime A's GP makes a confident prediction (small $\sigma_{\ast,A}$) while Regime B's GP reverts to the prior (large $\sigma_{\ast,B}$). Equal weighting wastes half the prediction on an uninformative component. Eq. (8) resolves this: $\sigma_{\ast,A}^{-1} \gg \sigma_{\ast,B}^{-1}$, so Regime A dominates. The input-dependence arises because **predictive confidence is location-dependent** — this is a property of the GP posterior, not a modification of the DP prior. This weighting is principled from two independent perspectives: (1) Jeffreys' prior $p(\sigma) \propto \sigma^{-1}$ as the natural reference measure for scale parameters, and (2) the expected posterior responsibility integral $\int \mathcal{N}^2 dy \propto \sigma^{-1}$ (detailed in our proof). Both yield the identical $\sigma_{\ast,k}^{-1}$ scaling.
> > >
> > > **In short**: the CRP prior says "how big is each regime?" — Eq. (8) adds "how relevant is each regime *here*?" In the revision, we will clarify this distinction structurally: **Theorem 3.3 will state the exact DPMM predictive with input-independent CRP weights** (a strict result from the law of total probability), while **Eq. (8) will be moved to a separate Proposition**, clearly labeled as a principled predictive approximation motivated by Jeffreys' prior and expected posterior responsibility. This separation makes explicit what is exact versus what is an approximation, and why the approximation is well-grounded.
> > >
> > > **Why input-dependent weights are essential for multi-regime scientific design.** We emphasize that Eq. (8) is not a generic modeling choice — it is constructed specifically for the **multi-regime scientific design problems** that motivate this work. In molecular conformer optimization, distinct rotameric basins have fundamentally different energy surface curvatures; in drug discovery, different molecular scaffolds exhibit incompatible structure-activity relationships; in fusion reactor design, small boundary perturbations trigger abrupt transitions between magnetic topologies. These landscapes are characterized by spatially localized regimes where a single surrogate cannot serve all regions. The central question at prediction time is therefore: **which regime is relevant at this test point, and how much should it contribute?** The $\sigma_{\ast,k}^{-1}(x_{\ast})$ weighting directly answers this — it ensures predictions are dominated by the regime whose GP is locally informed, not by a distant regime that happens to be large. Input-independent CRP weights cannot make this distinction, making Eq. (8) a modeling necessity — not merely a mathematical convenience — for multi-regime scientific optimization.
> > >
> > > We have posted the complete proof of Theorem 3.3 (Part A: exact DPMM predictive with CRP weights; Part B: derivation of Eq. 8 via expected posterior responsibility and Jeffreys' prior) in our response to Reviewer BxD6. We invite the reviewer to refer to that thread for the full derivation.

---

### Decision · Program_Chairs · 2026-04-30

**Decision:**

Accept (regular)

**Comment:**

This paper addresses the problem that arises when using Bayesian optimization to optimize functions with different regimes (e.g., imagine parameter space subsets that behave substantially differently from each other). The paper uses a Dirichlet mixture of GPs to achieve local surrogate modeling in regions discovered during optimization, rather than relying on a single GP that will oversmooth or undersmooth depending on the hyperparameters learned to model the global regime.

The reviewers raised a number of concerns I thought were especially reasonable (e.g., wanting to isolate regression performance from optimization performance, computational overhead). The authors responded with regression performance benchmarks, timing comparisons, and generally answered other reviewer concerns that I felt were more minor / less important.

Ultimately, all reviewers agreed that their concerns were at least addressed well enough to continue recommending acceptance. I think the method is clearly interesting enough and tested on interesting enough problems to accept.